# Identification of a cross-neutralizing antibody that targets the receptor binding site of H1N1 and H5N1 influenza viruses

Tingting Li [1,2,4], Junyu Chen[1,2,4], Qingbing Zheng [1,2,4], Wenhui Xue[1,2,4], Limin Zhang[1,2,4], Rui Rong[1,2], Sibo Zhang[1,2], Qian Wang[1,2], Minqing Hong[1,2], Yuyun Zhang[1,2], Lingyan Cui[1,2], Maozhou He [1,2], Zhen Lu[1,2], Zhenyong Zhang[1,2], Xin Chi[1,2], Jinjin Li[1,2], Yang Huang[1,2], Hong Wang[1,2], Jixian Tang[1,2], Dong Ying[1,2], Lizhi Zhou[1,2], Yingbin Wang[1,2], Hai Yu[1,2], Jun Zhang [1,2], Ying Gu [1,2] ✉, Yixin Chen [1,2] ✉, Shaowei Li [1,2] ✉ & Ningshao Xia [1,2,3] ✉

Influenza A viruses pose a significant threat globally each year, underscoring the need for a vaccine- or antiviral-based broad-protection strategy. Here, we describe a chimeric monoclonal antibody, C12H5, that offers neutralization against seasonal and pandemic H1N1 viruses, and cross-protection against some H5N1 viruses. Notably, C12H5 mAb offers broad neutralizing activity against H1N1 and H5N1 viruses by controlling virus entry and egress, and offers protection against H1N1 and H5N1 viral challenge in vivo. Through structural analyses, we show that C12H5 engages hemagglutinin (HA), the major surface glycoprotein on influenza, at a distinct epitope overlapping the receptor binding site and covering the 140-loop. We identified eight highly conserved (~90%) residues that are essential for broad H1N1 recognition, with evidence of tolerance for Asp or Glu at position 190; this site is a molecular determinant for human or avian host-specific recognition and this tolerance endows C12H5 with cross-neutralization potential. Our results could benefit the development of antiviral drugs and the design of broad-protection influenza vaccines.

Influenza is a respiratory infectious disease with a long-term threat to public health and substantial economic burden. Over the past five decades, seasonal influenza caused by influenza A viruses H1N1 and H3N2, and influenza B Yamagata and Victoria lineages, has led to the deaths of 250,000–500,000 people annually worldwide[1]. The influenza A virus is associated with higher morbidity and mortality rates than other influenza virus types and has a higher potential to cause a pandemic spread worldwide. Indeed, the historic 1918 Spanish H1N1 pandemic caused death to nearly one-third of the world's population (at least 40 million people)[2]. More recently, the outbreak of the novel pandemic influenza (H1N1) virus was first detected in Mexico in April 2009 and rapidly spread worldwide, with cases confirmed in over 200 countries. By August 6, 2010, the pandemic had resulted in 18,449 confirmed deaths[3]. Over the past decade, the virus associated with the 2009 H1N1 pandemic has remained a seasonal, annual hazard. Indeed, there have been serious pandemic threats from several highly pathogenic avian influenza viruses, such as H5N1, H5N6, and H7N9, which are also responsible for severe respiratory disease, with high mortality rates (~60%)[4]. Given the unpredictable nature of these viruses and their potential to spawn widespread disease, the most effective

[1]State Key Laboratory of Molecular Vaccinology and Molecular Diagnostics, School of Life Sciences, School of Public Health, Xiamen University, 361102 Xiamen, Fujian, China. [2]National Institute of Diagnostics and Vaccine Development in Infectious Diseases, Xiamen University, 361102 Xiamen, Fujian, China. [3]Research Unit of Frontier Technology of Structural Vaccinology, Chinese Academy of Medical Sciences, 361102 Xiamen, Fujian, China. [4]These authors contributed equally: Tingting Li, Junyu Chen, Qingbing Zheng, Wenhui Xue, Limin Zhang. ✉e-mail: guying@xmu.edu.cn; yxchen2008@xmu.edu.cn; shaowei@xmu.edu.cn; nsxia@xmu.edu.cn

countermeasure to combat influenza is a vaccine. Antiviral drugs, by comparison, have a narrow therapeutic window, with increased resistance and thus diminishing efficacy over time. However, despite vaccines being a better choice, influenza viruses vary considerably, and the current seasonal vaccine runs the risk of mismatch against newer, circulating strains. This therefore requires the development of next-generation vaccines that can elicit broad and/or cross-type protection against influenza.

Most broadly neutralizing antibodies (bnAbs) recognize hemagglutinin (HA), the major surface glycoprotein on influenza viruses[5]. HA mediates virus attachment to the host cell through the receptor-binding site (RBS) on the globular head region of HA1. The virus engages with sialic acid lining the cell surface, which leads to its subsequent pH-dependent entry through endosomal fusion. This fusion and entry is mediated by the fusion peptide on the HA stem region of HA2.

There are 18 antigenically distinct HA subtypes (H1-H18) of influenza A virus, generated as a result of rapid and continuous antigenic drift[6,7]. Heterosubtypic bnAbs can bind to multiple HA subtypes[8–14]. For instance, CR6261 antibody targets a highly conserved region in the HA stem domain and offers broad-protection against influenza A group 1 viruses[11]. In contrast, bnAb CR9114, which binds to a conserved epitope on the stem domain, can protect against lethal challenge with both influenza A and B viruses[15]. The successes in these studies offer a new strategy for preventing and treating influenza virus infection and provide hope for the development of a universal influenza vaccine.

Most of the cross-reactive antibodies produced to date map to the stem region of HA. However, there are a few cross-subtype nAbs that target the HA globular head, and these antibodies are usually more potent and immunodominant than stem antibodies; albeit, nAbs that target the HA globular head show a more limited neutralization

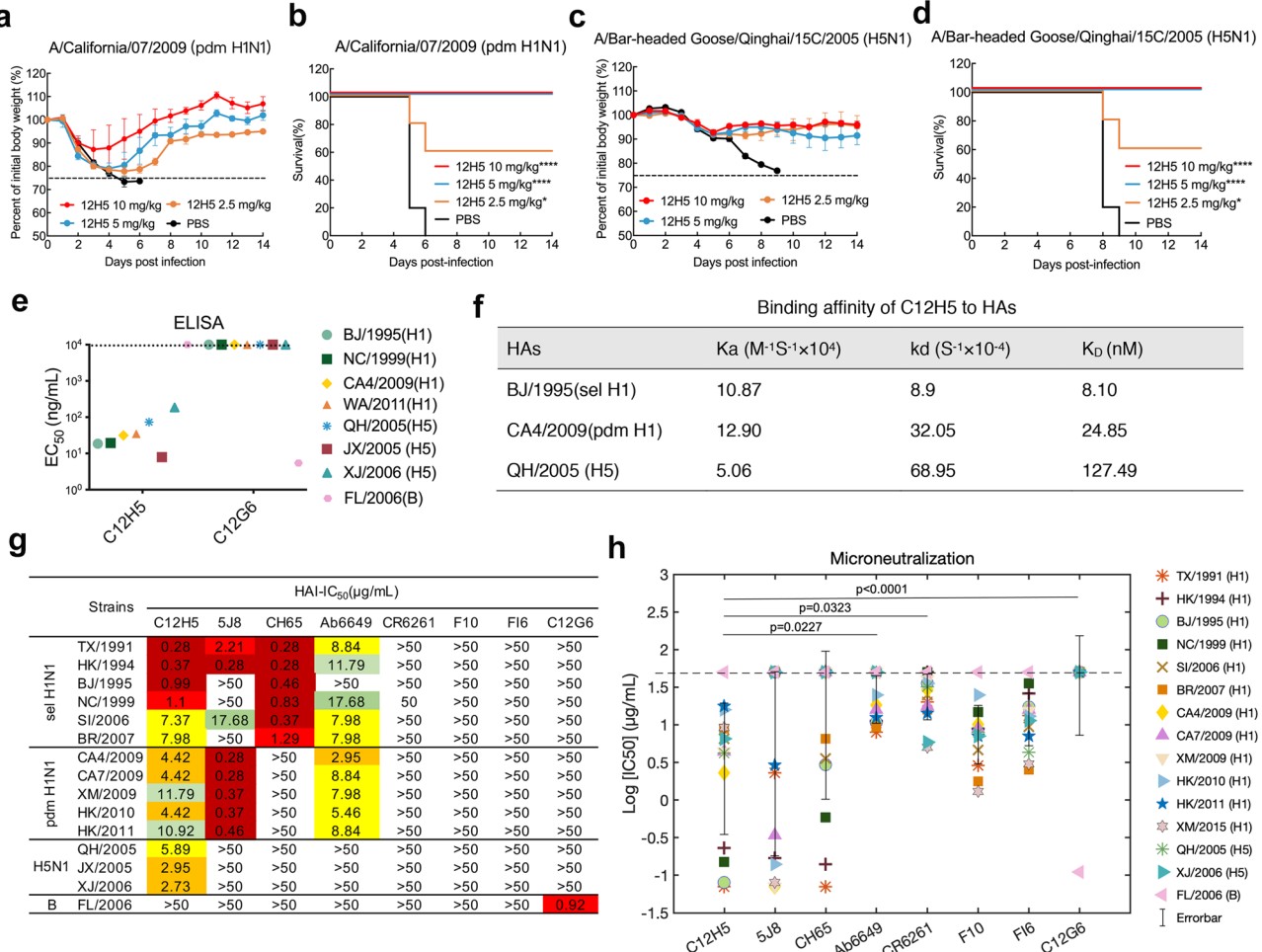

**Fig. 1 | In vivo therapeutic efficacy of 12H5 in mice and in vitro neutralization and binding activity of C12H5. a–d** Therapeutic efficacy of 12H5 against lethal challenge of MA-CA7/2009 (**a, b**), and MA-QH/2005 (**c, d**). The body weight (**a, c**) and survival curves (**b, d**) of BALB/c mice ($n = 5$ per group) treated with 12H5 or PBS 24 h after lethal challenge are shown. The body weight curves reflect the weights of surviving mice; data pertaining to the mice that died after challenge were omitted from the body weight data. Data in **a, c** are presented as mean values ± SD. For survival data, the log-rank test was used to evaluate significance, compared to control PBS-treated group (*$P < 0.05$, ****$P < 0.0001$). **e** Binding reactivity of C12H5 to purified HA proteins from H1N1 and H5N1 strains. $EC_{50}$ values above $10^4$ ng/mL were recorded as unfavorable. Full strain designations are detailed in Supplementary Table 2. **f** Binding affinity of C12H5 to HAs of three strains. The kinetic constants represent a representative experiment from two independent experiments. Data are presented as mean values. **g** 50% inhibitory concentrations [$IC_{50}$ (µg/mL)] were

determined by hemagglutinin inhibitory assay (HAI) against the representative stains from H1N1 and H5N1. Three independent experiments were used to calculate the average values. Values below 50 µg/mL are colored as follows: dark red, extreme reactivity; red, strong reactivity; orange, moderate-high reactivity; yellow, moderate-low reactivity; light green, weak reactivity; green, very weak reactivity; >50, negative reactivity. C12G6, the negative control antibody; FL/2006, control virus. **h** In vitro microneutralization ($IC_{50}$) values of C12H5 and the indicated antibodies against the 15 representative strains. Data represent the average values from three independent experiments and are marked with a single symbol. Data are presented as mean values +/− SEM. Group comparisons in **h** were made by Friedman test ANOVA with Dunn's multiple comparisons test. *$P < 0.05$ and ****$P < 0.0001$ were considered significant. Source data are provided as a Source Data file.

breadth. Specifically, C05[16], S139/1[17], and F045-092[18], which target the conserved epitope on the HA head region, are reported to be cross-reactive against H1, H2, and H3 subtypes. CH65[19], 5J8[20], and Ab6649[21], by comparison, are considered H1N1 bnAbs, none of which show cross-subtype reactivity. CH65 and 5J8 target the RBS, and 5J8 and Ab6649 have been shown to offer broadly neutralizing reactivity against seasonal and pandemic H1N1.

In this study, we isolated and characterized a broadly neutralizing H1N1 mAb−hereafter referred to as 12H5−and generated its human-mouse chimeric version, C12H5. We show that C12H5 neutralizes seasonal H1N1 (sel H1N1), the 2009 pandemic H1N1 (pdm H1N1), as well as the highly pathogenic avian influenza virus H5N1 in vitro. It also protects mice against H1N1 and H5N1 challenge in vivo. Through crystal structure and cryo-EM structure analyses of the 12H5 Fab in complex with the HA protein, we show that 12H5 targets the RBS conserved epitope. Using structure-guided mutagenesis, we identified eight highly conserved key residues involved in the neutralization. Our study suggests that C12H5 is a promising candidate for prophylactic and therapeutic countermeasures against H1N1 and H5N1 viruses. The cross-subtype neutralizing epitope provides a structural blueprint for the design of broadly protective vaccines against influenza A virus infection.

## Results

### An antibody 12H5 cross-neutralizes H1N1 and H5N1 viruses in vitro and in vivo

Mice were sequentially immunized intranasally or subcutaneously with live viruses of three representative strains of influenza A H1N1−A/Hong Kong/134801/1994 (HK/1994), A/New Caledonia/20/1999 (NC/1999), and A/Brisbane/59/2007 (BR/2007)−as immunogens for mAb production using standard hybridoma technology, as previously described[22] (see Methods). MAb 12H5 (IgG1) immediately stood out as an excellent antibody candidate in terms of seasonal H1N1 HA specificity, and was the only mAb to neutralize most of the tested viruses in all of the assays (see Methods).

We then tested the potential cross-reactivity of mAb 12H5 against several other influenza A virus strains (H1, H3, and H5) using hemagglutination inhibition (HAI) and microneutralization (MN) assays (Supplementary Table 1). We found that mAb 12H5 could neutralize nearly all annual H1N1 viruses (isolated years 1991 to 2015), including several vaccine strains; albeit the degree of neutralization varied amongst the different strains. Interestingly, mAb 12H5 also showed neutralizing activities against all tested 2009 pandemic H1N1 viruses and three avian H5N1 viruses, A/Chicken/Hong Kong/YU22/2002 (HK/2002, clade 8), A/Bar-headed Goose/Qinghai/15 C/2005 (QH/2005, clade 2) and A/Xinjiang/1/2006 (XJ/2006, clade 2) (Supplementary Table 1). Here again, 12H5 showed a wide range of neutralizing activity, with particularly high neutralization for H1N1 viruses isolated between years 1991 and 1999. Moreover, 12H5 showed great neutralization breadth; albeit, again, with different potencies against the different strains.

We then investigated the protective potential of mAb 12H5 against H1N1 and H5N1 virus infection in a mouse model. Groups of 6- to 8-week-old female mice ($n = 5$) were challenged with lethal doses of mouse-adapted (MA) influenza A viruses−MA-A/California/07/2009 (MA-CA7/2009, pdm H1N1) and MA-A/Bar-headed Goose/Qinghai/2005 (MA-QH/2005, H5N1)−and then treated with different doses of 12H5 or PBS (placebo) at 1 day after infection. The protection efficacy of 12H5 was dose-dependent: 5 mg/kg of 12H5 IgG protected 100% of mice from death and prevented substantial weight loss against lethal challenge with either MA-CA7/2009 (Fig. 1a, b) or MA-QH/2005 (Fig. 1c, d), whereas a lower dose (2.5 mg/kg) still afforded protection to some extent, with 50% to 60% survival rates (Fig. 1b, d). All of the PBS-treated control animals exhibited continuous weight loss following virus challenge and died within 10 days after infection. These results

indicate that passive immunization with mAb 12H5 provided hetero-subtypic protection against pandemic H1N1 and H5N1 viruses in mice.

The DNA sequences of the $V_L$ (variable region of immunoglobulin light chain) and $V_H$ (heavy chain) of 12H5 were acquired and aligned with the most adjacent germline sequence using the IMGT database (http://www.imgt.org). The result showed 12H5 aligns with a KV3-4 and HV9-1 germline V gene and its nucleotide identity is 98.28% (286/291 nt) for the $V_L$ chain and 95.83% (276/288 nt) for the $V_H$ chain (Supplementary Fig. 1). To evaluate the potential clinical application of 12H5, we constructed and carried out further characterizations using a chimeric 12H5 monoclonal antibody−designated C12H5−that contains the variable region of mouse 12H5 and the human IgG1 Fc region.

### C12H5 cross-neutralizes H1N1 and H5N1 viruses in vitro

To further profile the breadth of activity of C12H5 against influenza A viruses, we first tested C12H5 with recombinant HA proteins from two seasonal H1N1 strains, two pandemic H1N1 strains, three H5N1 strains, and one type B strain using ELISA. C12H5 showed high reactivities with all H1N1 and H5N1 HAs, with half-maximal effective concentration ($EC_{50}$) values ranging from 7.9 to 186.5 ng/mL (Fig. 1e). In contrast, the control antibody C12G6 (a broadly chimeric mAb against influenza B viruses from our previous study[23]) did not react with any of the influenza A viruses except for the B/Florida/4/2006 (FL/2006) control (Fig. 1e). Next, C12H5 binding was measured against three representative strains of HA using surface plasmon resonance (SPR). C12H5 showed strong affinities to sel A/Bejing/262/1995 (BJ/1995) HA and pdm A/California/04/2009 (CA4/2009) HA, with affinity constants ($K_D$) of 8.1 nM and 24.85 nM, respectively (Fig. 1f). Most notably, C12H5 showed heterosubtypic binding activity to H5 QH/2005 HA with a $K_D$ of 127 nM (Fig. 1f), indicating a moderate interaction but one that is within the range that has been reported to effectively neutralize the virus ($K_D < 250$ nM) (Fig. 1f)[17], while its relatively fast off rate might restrict its effectiveness and potency in some assays. Consistent with its reactivities to H1N1 and H5N1 viruses, C12H5 also showed binding to Madin-Darby canine kidney (MDCK) cells pre-infected with sel H1N1 NC/1999, CA4/2009, and QH/2005 viruses in an immunofluorescence assay (Supplementary Fig. 2).

We then compared C12H5 with six previously reported subtype-specific or cross-subtype neutralizing influenza A HA-specific bnAbs: 5J8[20], CH65[19], Ab6649[21], CR6261[11], F10[12], and FI6[24]. First, we generated chimeric forms of the antibodies, as done for C12H5, with all chimeric versions demonstrating good binding activity against H1 or H5 virus HA, consistent with the original reports (Supplementary Fig. 3 and Table 2). Using these chimeras, we showed that the 5J8 antibody offered HAI activities against most of the sel H1 and all of the pdm H1 strains, but not against NC/1999, BJ/1995, BR/2007, or three H5N1 viruses. Of all the antibodies, CH65 displayed the most robust HAI activity but only to the seasonal H1 strains. 5J8 also showed just as robust activities to the pdm H1N1 strains, CH65 and 5J8 both did better against many viruses compared to C12H5 within these subgroups (Fig. 1g). Comparatively, Ab6649 exhibited moderate HAI activity to sel H1N1 and pdm H1N1 but not to H5N1 viruses. And, finally, none of CR6261, F10, or FI6 showed HAI activity against any of the influenza A viruses, most likely due to their stem specificity. In contrast, our chimeric C12H5, however, showed high HAI activity against all major seasonal H1N1 viruses except for SI/2006 and BR/2007, and moderate HAI activities against pandemic H1N1 and three H5N1 strains (Fig. 1g).

In the MN assay, C12H5, CR6261, F10, and FI6 antibodies neutralized both H1 and H5 viruses with varied potency (Fig. 1h and Supplementary Fig. 4), whereas 5J8, CH65, Ab6649, and CR6261 were unable to neutralize some strains. Overall, the mean log $IC_{50}$ value for C12H5 against the panel of virus strains is not significantly different to that of 5J8 and CH65; however, unlike the broad neutralization breadth

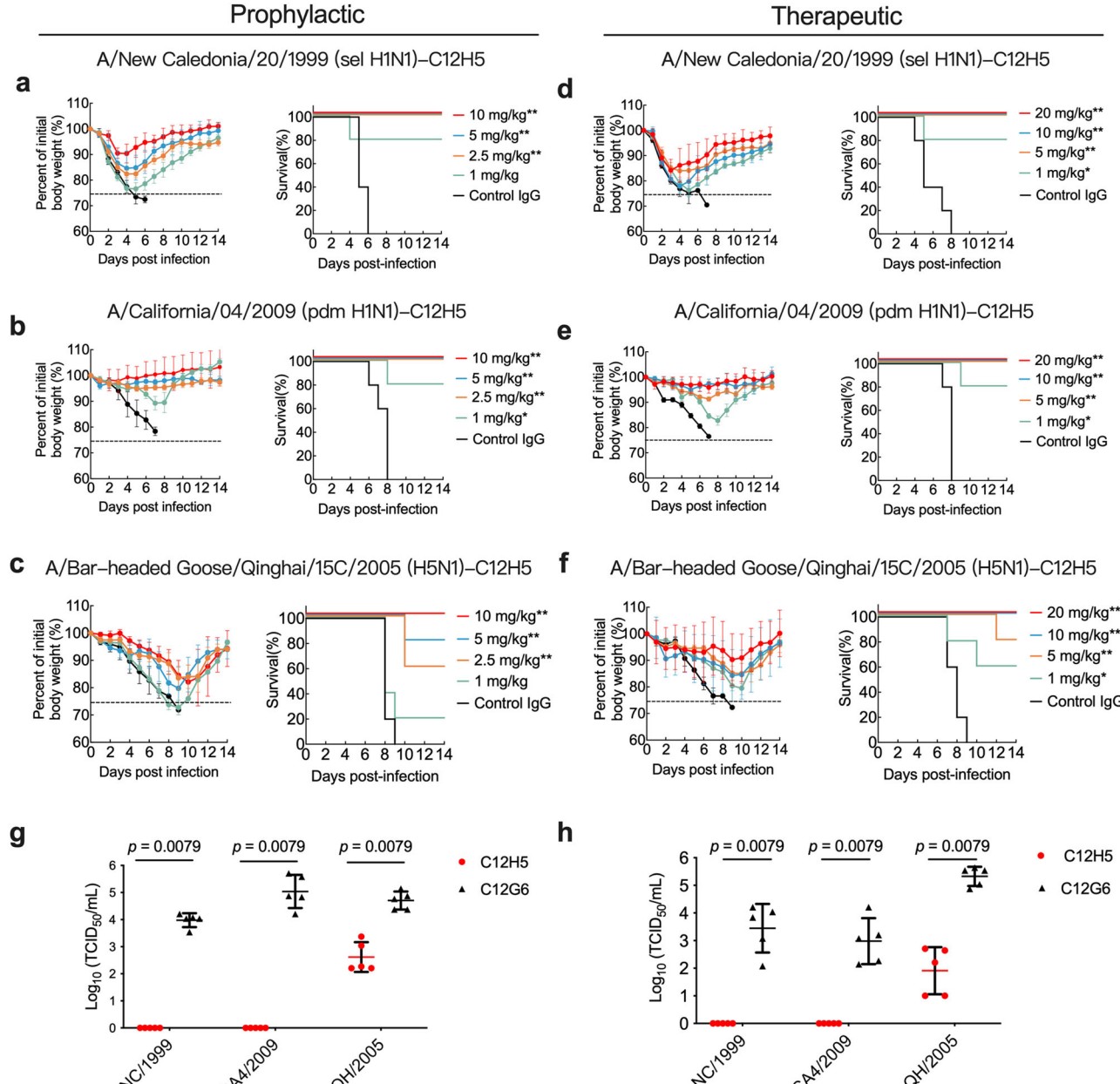

**Fig. 2 | In vivo prophylactic and therapeutic efficacy of C12H5 in mice. a** to **c** Prophylactic efficacy of C12H5 against lethal challenge of MA-NC/1999 (**a**), MA-CA4/2009 (**b**), or MA-QH/2005 (**c**). Body weight (left) and survival (right) curves of BALB/c mice (*n* = 5 per group) treated with 10, 5, 2.5, or 1 mg/kg C12H5, or C12G6 (10 mg/kg) 24 h before lethal challenge are shown. **d** to **f** Therapeutic efficacy of C12H5 against lethal challenge with MA-NC/1999 (**d**), MA-CA4/2009 (**e**), and MA-QH/2005 (**f**). Body weight (left) and survival (right) curves of BALB/c mice (*n* = 5 per group) treated with 20, 10, 5, or 1 mg/kg C12H5, or C12G6 control (20 mg/kg) 24 h after lethal challenge are shown. The body weight curves in **a**–**f** reflect the weights of surviving mice; data pertaining to the mice that died after challenge were

omitted from the body weight data. **g** Lung virus titers from mice treated with C12H5 (10 mg/kg) or C12G6 control (10 mg/kg) in the prophylactic group (*n* = 5 per group) were determined 6 days after infection. **h** Lung virus titers from mice treated with C12H5 (20 mg/kg) or C12G6 control (20 mg/kg) in the therapeutic group (*n* = 5 per group) were determined 6 days after infection. Data in **a**–**h** are presented as mean values ± SD. For survival data, the log-rank test was used to evaluate significance. For virus titers, each group was compared with the control group using a two-tailed Mann–Whitney *U*-test (***$p < 0.001$; **$p < 0.01$; *$p < 0.05$). $TCID_{50}$, median tissue culture infective dose. Source data are provided as a Source Data file.

of C12H5, 5J8 was unable to neutralize four of the seasonal H1N1 strains and the two H5N1 strains, and CH65 was incapable of neutralizing all of the tested pdm H1N1 strains and the two H5N1 strains (Fig. 1h and Supplementary Fig. 4). On the other hand, the mean log IC50 value for C12H5 is significantly lower than that for Ab6649, CR6261 and the control antibody, C12G6, suggesting that C12H5 is more potent than Ab6649 and CR6261 at least in this assay. Meanwhile, C12H5 exhibited broad-spectrum neutralization activity against all representative H1 strains, with a median $IC_{50}$ of 5.75 μg/mL, as compared with that for

F10 and FI6 (7.72 and 13.45 μg/mL, respectively) (Supplementary Fig. 4). Moreover, C12H5 exclusively offered cross-neutralization against the H5N1 virus; none of other H1 head-region antibodies recognized H5 (Fig. 1h and Supplementary Fig. 4). Taken together, despite the varied neutralizing potency among the different strains, we show that C12H5 outperforms the other three reported H1 head-region bnAbs in terms of neutralization breadth and has an overall higher neutralization potency than the three stem bnAbs tested (CR6261, F10, FI6).

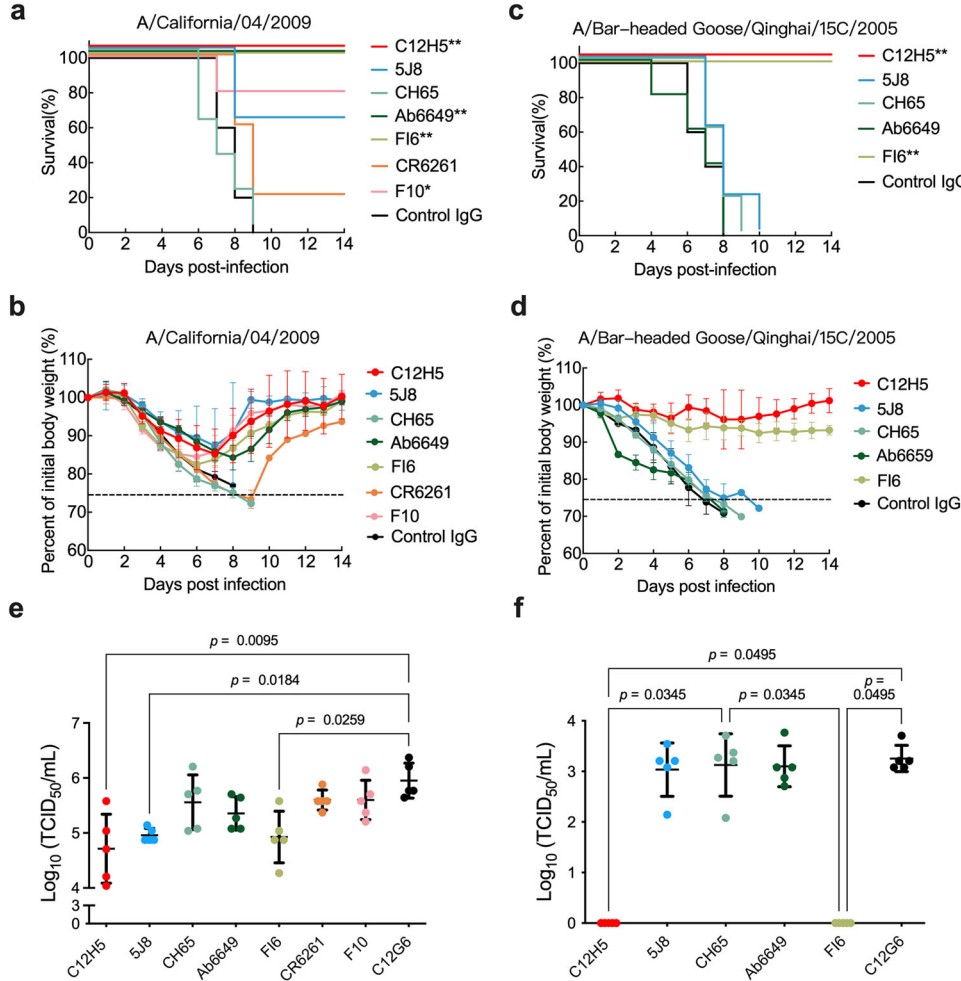

**Fig. 3 | Comparison of therapeutic efficacies of C12H5 and other bnAbs in mice.** Survival curves (**a**, **c**), body weight changes (**b**, **d**), and lung virus titers (**e**, **f**) for BALB/c mice ($n = 5$ per group) treated intravenously with antibodies (3 mg/kg for MA-CA4/2009 virus, 15 mg/kg for MA-QH/2005 virus) 24 h after lethal challenge. The body weight curves reflect the weights of surviving mice; data pertaining to the mice that died after challenge were omitted from the body weight data. Virus titers in the lungs were determined 6 days after infection. Data in **b**, **d**, **e**, **f** are presented as mean values ± SD. For **a**, **c**, statistical analyses were performed using the log-rank test. For **e**, **f**, statistical analyses were performed using Kruskal–Wallis with Dunn's multiple comparisons test (***$p < 0.001$; **$p < 0.01$; *$p < 0.05$). Source data are provided as a Source Data file.

## C12H5 shows cross-type prophylactic and therapeutic activities against H1N1 and H5N1 viruses in mice

We next evaluated the prophylactic and therapeutic efficacy of C12H5 against the MA influenza A viruses—MA-A/New Caledonia/20/1999 (H1N1, MA-NC/1999), MA-CA4/2009, and MA-QH/2005—using different doses delivered by intravenous administration. Antibodies were administered 24 h before and after virus infection for prophylactic and therapeutic groups, respectively.

In the prophylactic groups ($n = 5$), doses of 2.5 mg/kg and above could fully protect mice from lethal infection with sel H1N1 MA-NC/1999 and pdm H1N1 MA-CA4/2009 virus (Fig. 2a, b), with these doses also having little effect on body weight for mice in the pdm H1N1 group (Fig. 2b). Doses as low as 1 mg/kg still offered 80% survival in both H1N1 groups (Fig. 2a, b). For the highly pathogenic H5N1 virus, a dose of 10 mg/kg completely protected mice against lethal infection with MA-QH/2005 virus (Fig. 2c), whereas a lower 2.5 mg/kg dose offered partial protection, with a survival rate of 60%.

In the therapeutic groups ($n = 5$), 5 mg/kg and above could fully protect mice against lethal challenge with MA-NC/1999 virus; although, body weight did fluctuate but recovered during the experimental period (Fig. 2d). Mice were fully protected against a lethal infection of MA-CA4/2009 with 5 mg/kg or higher dosages, with less changes in body weight observed compared to that of MA-NC/1999

(Fig. 2e). In addition, a dose of 1 mg/kg still provided 80% protection against the virus (Fig. 2e). For the H5 virus, doses of 10 mg/kg and above effectively alleviated the weight loss observed and completely protected mice against MA-QH/2005 virus infection (Fig. 2f). The body weight loss is shifted by several days compared to that MA-NC/1999, which might be due to resulting pathogenicity varying even at the same dose of antibodies across different virus strains. Mouse survival rates were 80% with 5 mg/kg, and 60% with 1 mg/kg (Fig. 2f). Consistent with the survival data, lung viral titers were determined 6 days after virus infection and were considerably reduced in mice administered with C12H5 as compared with the control (IgG) group (Fig. 2g, h). The prophylactic and therapeutic experiments collectively indicated that C12H5 could protect mice against H1N1 and H5N1 viruses in vivo, despite less effectively to H5.

To further investigate the potency of C12H5, we compared its therapeutic efficacy against other representative antibodies at two doses. We found that the lower dose of 3 mg/kg of C12H5, Ab6649, or FI6 fully protected mice from lethal challenge with MA-CA4/2009 (Fig. 3a and Supplementary Fig. 5). In contrast, 3 mg/kg of F10, 5J8, CH65 or CR6261 provided only partial or no protection against infection, with survival rates of 80%, 60%, 0%, and 20%, respectively (Fig. 3a). The potent protection achieved with C12H5, Ab6649, and FI6 was also reflected in body weight

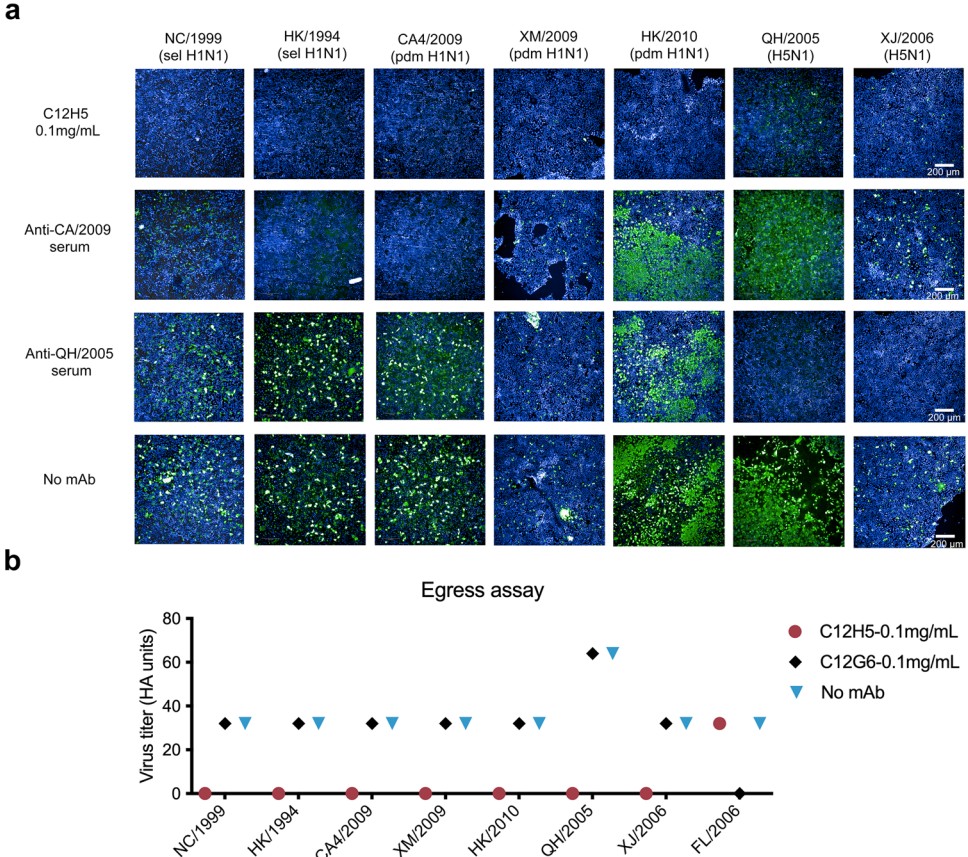

**Fig. 4 | Neutralization mechanism of C12H5. a** MDCK cells were inoculated with NC/1999, HK/1994, CA4/2009, XM/2009, HK/2010, QH/2005 or XJ/2006 viruses pre-incubated with C12H5 or polyclonal rabbit sera. The expression of influenza A virus nucleoprotein in MDCK cell monolayers at 16 to 18 h after inoculation was detected by immunofluorescence. Green, infected cells positive for NP protein; blue, 4′,6-diamidino-2-phenylindole staining. **b** Egress assay of influenza A virus detected in the supernatants of MDCK cells infected with NC/1999, HK/1994, CA4/2009, XM/2009, HK/2010, QH/2005, XJ/2006 or FL/2006 and subsequently incubated with different concentrations of C12H5, as indicated. Influenza A viruses were determined by Hemagglutinin assay, with the data represented as HA units. Source data are provided as a Source Data file.

variation, with much less fluctuation in body weight curves for mice in these three groups as compared with mice treated with the other antibodies (Fig. 3b). Among the mice challenged with a lethal infection of H5N1 MA-QH/2005 virus, only those treated with C12H5 and FI6 at a dose of 15 mg/kg were fully protected, with no weight loss. Mice treated with 5J8, CH65, and Ab6649 at the same high dose all died within 2 weeks post-virus challenge (Fig. 3c, d). Consistent with the survival and body weight data, the lung virus titers for MA-CA4/2009 (pdm H1N1) and MA-QH/2005 (H5N1) were considerably lower in mice treated with C12H5 and FI6 than in those treated with the other bnAbs. Mice treated with 5J8 showed a relatively low survival rate of 60%, despite of low virus titers in the lung following MA-CA4/2009 (pdm H1N1) challenge (Fig. 3e, f).

### Neutralization mechanisms of mAb C12H5
The results above showed that C12H5 offered excellent HAI and MN activities against H1N1 and H5N1 viruses. Thus, we further explored the neutralization mechanism mediated by C12H5. First, we tested whether C12H5 functions by inhibiting virus entry into MDCK cells, as determined using an immunofluorescence assay. Consistent with the HAI results, C12H5 provided inhibition against sel H1N1(NC/1999, HK/1994) and pdm H1N1 (CA4/2009, XM/2009, HK/2010) virus entry into MDCK cells (Fig. 4a). The inhibition activity against two H5N1 viruses (QH/2005, XJ/2006) was relatively weaker than that for the H1N1 virus, and this might be due to its moderate affinity to the H5N1 virus (Figs. 4a and 1f).

Next, we determined whether C12H5 inhibits influenza A virus via an alternative mechanism, such as by blocking virus egress. We carried out virus egress inhibition assays and collected the cell supernatant to determine virus titers using a hemagglutinin assay[25]. C12H5 at a concentration of 0.1 mg/mL inhibits the egress of all above-mentioned viruses from the infected cells except for FL/2006 strain. The control antibody C12G6 also showed no inhibition of the egress activities of H1N1 or H5N1 viruses (Fig. 4b). Thus, C12H5 exhibits similar neutralization mechanisms against the tested viruses. Specifically, we show that C12H5 exerts its antiviral activity against two seasonal and three pandemic H1N1 strains and—to a lesser extent—two H5N1 strains, by directly inhibiting the virus from attaching to the cell. Furthermore, C12H5 can effectively block virus egress, as tested with five H1 and two H5 strains but FL/2006 strain.

### Structural characterization of 12H5 in complex with HA
To elucidate the cross-subtype neutralization function of 12H5, we prepared and resolved the crystal structure of 12H5 Fab (mouse antibody) in complex with the HA of A/California/04/2009 (H1N1) to a resolution of 3.1 Å, which will be referred to as 12H5: HAhr$^{CA}$ (head region of CA4/2009 HA) (Fig. 5a; Supplementary Fig. 6 and Table 3). The initial crystal sample contained a well-characterized trimer HA and Fab complex (Supplementary Fig. 7a); however, two degraded HA monomer head regions (aa 56-263; H3 numbering), each bound by a 12H5 Fab, were present in the asymmetric unit of complex. We surmise that the long-term crystallization process of ~2 months may have led to the degradation of the HA trimer (Supplementary Fig. 6e), as also

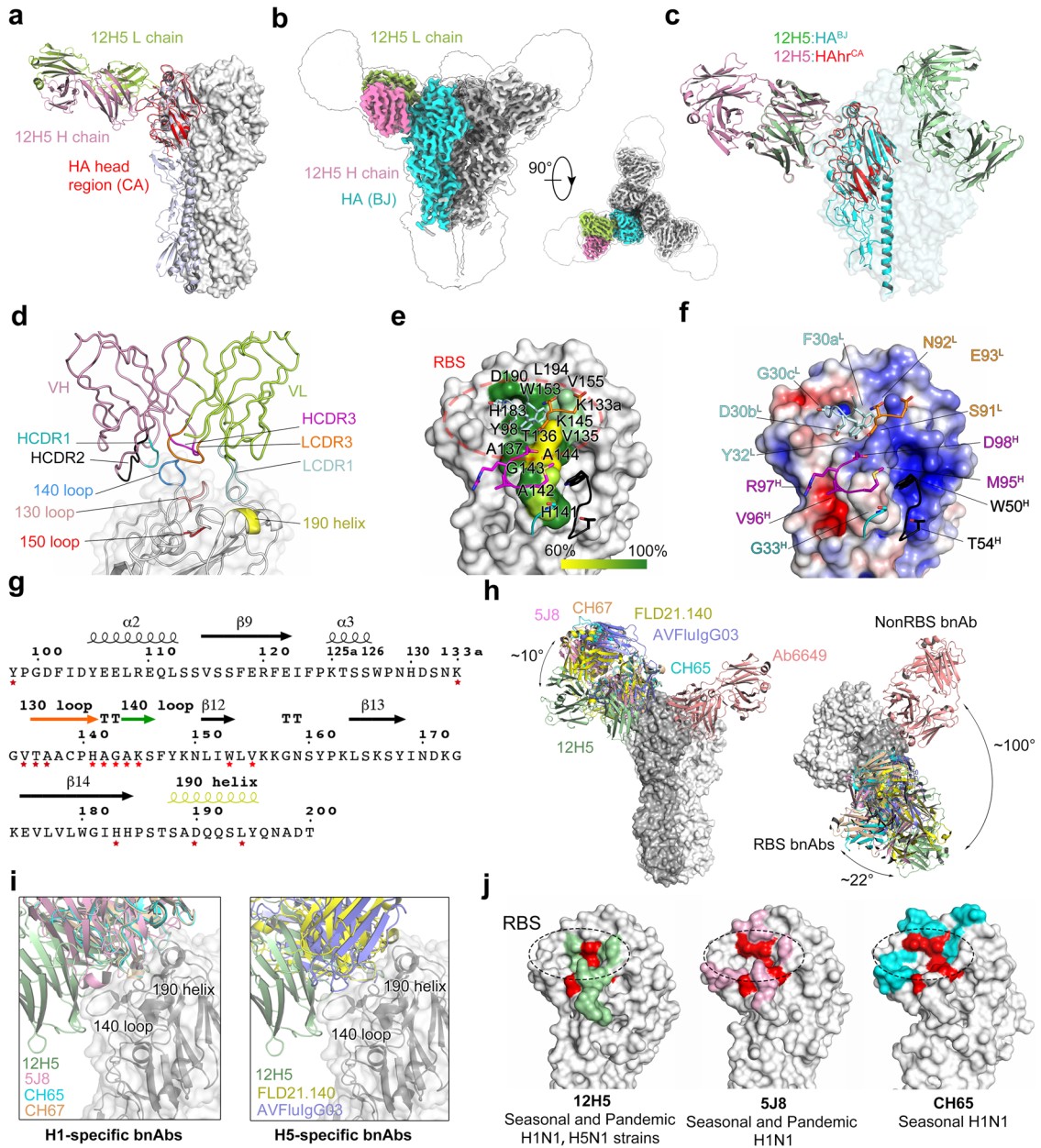

**Fig. 5 | Structure of 12H5:H1 HA complex. a** The crystal structure of 12H5 in complex with HA of CA4/2009 (12H5 HAhr$^{CA}$ complex). The CA HA head region (i.e., HAhr$^{CA}$) of the 12H5:HAhr$^{CA}$ complex was superimposed onto one protomer of a CA HA trimer (PDB code: 3lzg). The HA head region is in red, one protomer of the HA trimer is in light purple; the light chain and heavy chain of 12H5 are in light green and pink, respectively. **b** The overall Cryo-EM structure of 12H5 in complex with HA trimer from BJ/1995 (12H5:HA$^{BJ}$) at 3.15 Å resolution in side view and top view. **c** The 12H5:HA$^{BJ}$ structure allowed the exact fit of the crystal structure of 12H5: HAhr$^{CA}$ complex. **d** to **f** Interaction analysis of 12H5: HAhr$^{CA}$ complex. **d** Close-up view of the interface of the 12H5: HAhr$^{CA}$ complex. **e** Sequence conservation of the 12H5 epitope. HA is in surface representation, and the conservation of the epitope residues in H1N1 subtypes are color-coded on the surface. **f** Electrostatic potential surface of

HA is illustrated in gradient color: red, negative, −4KT; blue, positive, +4KT, white, neutral. The side chain or main chain of 12H5 that makes contact with residues on HA are depicted in stick mode. **g** The sequence of the receptor-binding site (RBS) domain. Residues at the interface are marked with red asterisks. **h** Comparison of the binding orientation of 12H5 with other H1 head antibodies:12H5 (green), 5J8 (pink, PDB code: 4M5Z), CH65 (cyan, PDB code: 5UGY), CH67 (Orange, PDB code: 4HKX), Ab6649 (salmon, PDB code: 5W6G), FLD21.140 (yellow, PDB code: 6A67) and AVFluIgG03 (purple, PDB code: 5DUP). The right panel view is horizontally rotated 90° with respect to the left panel. **i** Close-up view of the interaction between H1-specific or H5-specific antibodies and HA. **j** The footprint of the three H1N1 head antibodies. The unique epitopes of 12H5, 5J8 and CH65 are colored in green, pink, and cyan, respectively. Overlapping regions are in red.

observed in our previous study[26]. Nevertheless, the structure of the HA head region was congruent with the corresponding moiety of the CA4/2009 HA trimer structure[27] (PDB code: 3lzg), as manifested by a structural superimposition of a root mean square deviation (RMSD) of 0.5 Å for all HA1 C-α atoms. We showed that the 12H5 epitope locates to the receptor-binding region (Fig. 5a).

We also solved the cryo-EM structure of 12H5 with the HA trimer of A/Beijing/262/1995 (H1N1) (12H5: HA$^{BJ}$; HA trimer of BJ/1995 strain) at a resolution of 3.14 Å (Fig. 5b, Supplementary Fig. 8 and Supplementary Table 4). Fitting the 12H5: HAhr$^{CA}$ into the model of 12H5: HA$^{BJ}$ revealed congruency in the Fab binding orientations in the monomer HA$^{CA}$ head region and trimer HA$^{BJ}$ (Fig. 5c); the cryo-EM structure also

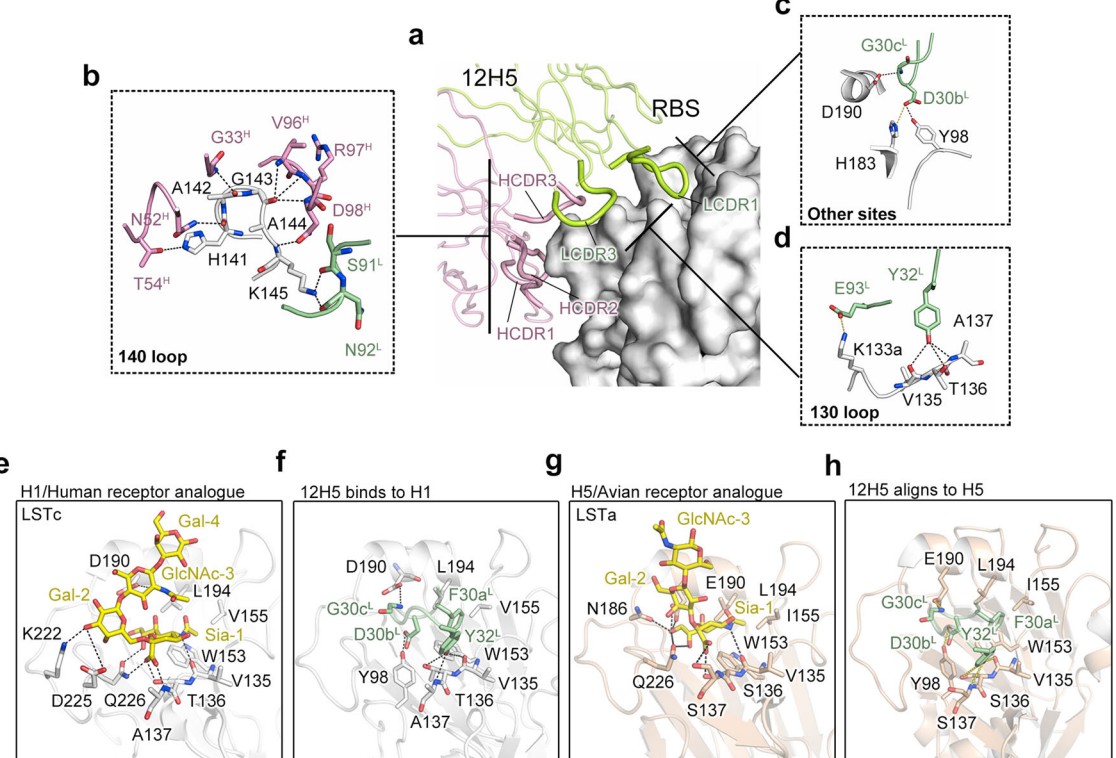

**Fig. 6 | 12H5 mimics the receptor-binding mode of H1 and H5. a–d** Hydrogen bonds and salt bridges between 12H5 and 140-loop (**b**), 130-loop (**d**) and other sites (D190, H183 and Y98) (**c**). **e–h** Receptor mimicry of 12H5. **e** Hydrogen bond interactions between the receptor analog LSTc and the H1 RBS, as shown in the co-crystal structure (PDB code: 3UBE). The carbohydrate group of the LSTc Sia terminus stretches into a hydrophobic cavity enfolded by side chains of W153, L194, and V155. **f** Hydrogen bonding between the 12H5 LCDR1 tip and the V135-A137 stretch. Hydrogen bonds are shown as black dashed lines. F30a^L of LCDR1 mimics the hydrophobic interaction with the RBS, resembling the carboxylate group of the LSTc Sia terminus. **g** Hydrogen bonds and hydrophobic interactions between the receptor analog LSTa and H5, as shown in the co-crystal structure (PDB code: 4K63). **h** 12H5: HAhr^CA complex is aligning to the H5 structure. The probable hydrogen bonds are shown as yellow dashed lines.

confirmed that 12H5 indeed targets the RBS of different protomers within the same HA and could occupy all three potential binding sites.

In the crystal structure of 12H5: HAhr^CA, 12H5 interacted with the HA head region, creating a buried surface area of 827 Å² on HA1, as calculated by PISA (Supplementary Table 5). The antibody heavy chain and light chain mediate 52% and 48% of 12H5:HAhr^CA contacts, respectively (Fig. 5d and Supplementary Table 5). 12H5 primarily uses its heavy chain complementarity-determining region (HCDR)−1, HCDR2, and HCDR3 to contact HA, along with the epitope center on the negatively charged 140-loop. Its light chain (L)-CDR1, on the other hand, interacts with the negatively charged RBS groove, with contact formed primarily by the 130-loop, the 150 loop, and the 190 helix. Finally, LCDR3 makes some connections with the edge of the RBS (Fig. 5d–f). The CDRs recognition defines a non-continuous conformational epitope, mostly focusing on 130- and 140-loop stretches of residues; the sequence and secondary structure of the epitope are depicted in Fig. 5g. The epitope of 12H5 is highly conserved (89.2%) among all H1N1 isolates (*n* = 8907) (Fig. 5e and Supplementary Table 6). Compared with other H1-specific (5J8, CH65, CH67[28], and Ab6649) or H5-specific (FLD21.140[29] and AVFluIgG03[30]) RBS antibodies, 12H5 targets the HA RBS ~10° lower than 5J8, CH65, CH67, FLD21.140, and AVFluIgG03 when viewed along the HA trimer's longitudinal axis, and with an ~100° rotation clockwise from Ab6649 when viewed from the top (Fig. 5h). Among these antibodies, only 12H5 covers the 140-loop (Fig. 5i). Thus, the footprint of 12H5 (827 Å²) is more membrane-proximal than 5J8 and CH65. As such, there is a 280 Å² overlapping area across the footprints of these three antibodies (Fig. 5j). As compared with 5J8, which is

most similar, C12H5 has distinctive epitope-based amino acids located within the 130- and 140-loops (Fig. 5j).

## 12H5 mimics the receptor-binding mode of H1 and H5

In the 12H5: HAhr^CA structure, 12H5 makes extensive interactions with HA, creating 15 hydrogen bonds and two salt bridges, mostly with the 140-loop, 130-loop and 190 helix (Fig. 6a and Supplementary Table 5). The HCDR2 and HCDR3 make 7 hydrogen bond contacts with the H141-K145 stretch of residues, whereas LCDR3 makes two hydrogen bond contacts with K145. Collectively, these interactions account for 60% of the total hydrogen bonds (Fig. 6b). Residues H141, A142, G143, and K145 make hydrogen bonds with the heavy chain and light chain. G30c^L uses its main chain to interact with the side chain of the conserved D190, and D30b^L uses its side chain to interact with another two conserved residues, H183 and Y98, located on the floor of the RBS (Fig. 6c).

Interestingly, we found that LCDR1 of 12H5 deeply extends to the groove of HA1 RBS, which structurally mimics the interaction between H1 HA and human receptor LSTa (α 2, 6) pentasaccharide (PDB code: 3ube)[31] (Fig. 6e, f). In LCDR1, the hydroxy moiety of the Y32^L side chain interacts with receptor-binding site V135, T136, and A137, mediated via three hydrogen bonds; this interaction mimics the interaction of Sia-1 with the 130-loop to some extent. (Fig. 6d–f). In the 190 helix, the main chain of D30b^L interacts with D190 by hydrogen bonding, similar to the interaction between GalNac-3 and D190 (Fig. 6d–f). Meanwhile, F30a^L mediates hydrophobic interactions with the highly conserved W153, V155, and L194 residues, recapitulating the receptor-binding mode (Fig. 6e, f).

We then aligned the 12H5 complex to the H5 receptor complex structure[32] (PDB code: 4K63) and found the 12H5 LCDR1 fits well to the

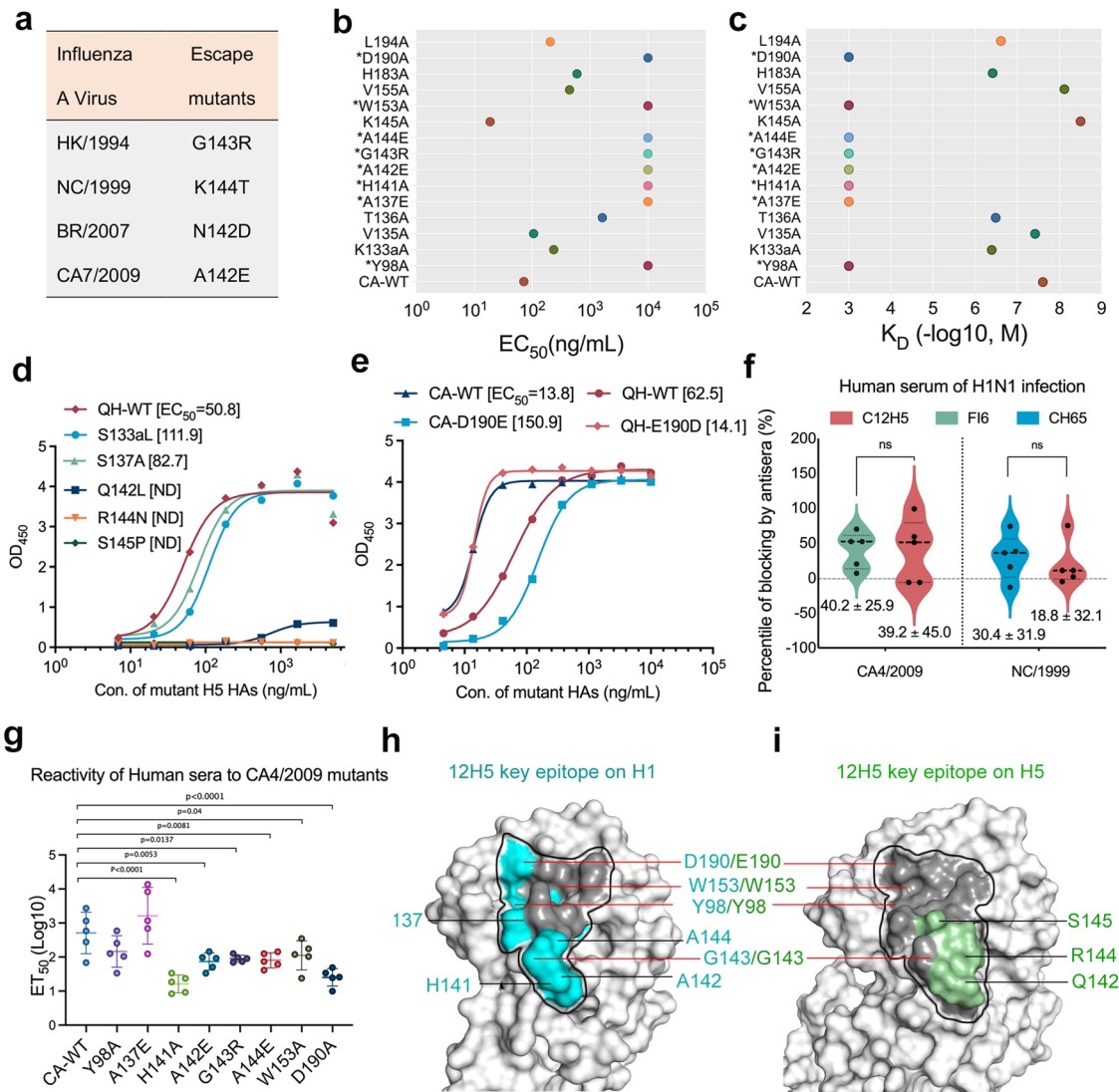

**Fig. 7 | Interaction analysis and mutagenesis for the epitope of 12H5. a** Amino acid substitutions found in the HA of C12H5-induced escape mutants with four influenza A strains. **b** Reactivity profiles of CA HA and its mutants against C12H5 antibodies measured through sandwich ELISA. The EC₅₀ values were calculated by sigmoid fitting, as shown in Supplementary Fig. 11. EC₅₀ values are plotted as circles along the horizontal axis for each protein. **c** Binding affinity measurements for HA and its mutants against C12H5, as determined with SPR. The kinetic constants between C12H5 and Y98A, A137E, H141A, A142E, G143R, A144E, W153A, and D190A mutants were not determinable. The calculated affinity constants and fitting results are shown in Supplementary Table 8 and Fig. 11. The EC₅₀ and $K_D$ values in **b** and **c** are averaged data from two independent experiments. **d** Comparison of binding reactivity of five variant sites with the mutants and wild type of QH/2005 HA in two

experiments. ND, not detectable. **e** Comparison of binding reactivity of 12H5 to the 190 mutants and wild type of CA4/2009 and QH/2005 HA from two experiments. **f** Blocking assay by human sera ($n = 5$) of H1N1 infection (H1N1-RNA positive) to antibodies binding. Two groups were compared with the control group using a paired two-tailed $t$-test. NS means not statistically significant. **g** ELISA assay to analysis the sensitivity of human sera ($n = 5$) of H1N1 infection to the critical point mutations for C12H5 recognition. Statistical analyses were performed using one-way ANOVA with Dunn's multiple comparisons test (*$P < 0.05$, **$P < 0.01$ and ****$P < 0.0001$). **h, i** Critical residues of 12H5 on HA of H1N1 and H5N1 viruses; the outlined black region shows all epitopes of 12H5; the same amino acid types are indicated with red lines. Data in **b**–**e** are presented as mean values, data in **f, g** are presented as mean values ± SD. Source data are provided as a Source Data file.

H5 RBS. Compared with H1, the RBS of H5 has different side chain conformations at some substitutions, such as T136S, A137S, V155I, and D190E. However, 12H5 also mimics the binding mode of H5 with the avian receptor LSTc (α 2, 3) pentasaccharide by forming favorable hydrogen bonds with residues V135-S137 and especially E190, as well as mediating hydrophobic interactions with residues W153, I155, and L194 (Fig. 6g, h). The mimicry of both H1 and H5 receptor-binding modes thus contributes to the cross-neutralizing reactivity that we see for C12H5. Our structure analysis showed that the interaction between the main chain of C12H5 G30c^L and the side chain of HA1 D190 (an H1-specific site and related to host transition from human to avian, E190 for H5 instead) forms a favorable hydrogen bond. To check if E190 is also favored for the G30c^L interaction, we superimposed a H5 HA

structure bearing E190 residue (PDB code: 3UBE) onto the 12H5: HAhr^CA immune complex structure. We found that the side chain of E190 would similarly create hydrogen bonds with the main chain of C12H5 G30c^L like D190 (Supplementary Fig. 9). This structurally explains how C12H5 accommodates the D→E substitution at position 190 between H1 and H5.

## Sequence analysis and structure-based mutagenesis of 12H5 epitope

To further investigate the molecular basis of binding and to identify the binding domain of C12H5, we generated C12H5-induced escape mutants of influenza H1N1 by culturing the viruses in the presence of C12H5. We identified four escape mutants: G143R in HK/1994 strain,

K144T in NC/1999 strain, N142D in BR/2007 strain, and A142E in CA7/2009 strain (Fig. 7a and Supplementary Table 7). All of the mutated residues are located at or near the HA RBS of H1N1 (Fig. 5e). To ascertain the critical residues for C12H5 binding, alanine-scanning mutagenesis (Supplementary Fig. 10a) were carried out along the 15 residues of HA that were involved in mediating interactions with 12H5 in the co-crystal structure, and these mutations were checked by variations in $EC_{50}$ and $K_D$ values. In the case that the residue in question was Ala, its contribution was checked by substitution with Glu (with a long side chain). Eight mutants showed dramatically lower binding reactivities to 12H5: Y98A, A137E, H141A, A142E, G143R, A144E, W153A, D190A; for these mutants, the $EC_{50}$ values were more than 100 times higher than that for WT HA (Fig. 7b and Supplementary Fig. 11). SPR responses to C12H5 were also undetectable (Fig. 7c and Supplementary Table 8), consistent with the ELISA and escape mutant results. In contrast, point mutations at K133a, V135, T136, K145, V155, H183, and L194 had no significant effect on antibody binding. Interestingly, the K145A mutant showed significantly enhanced binding affinity to C12H5, with a $K_D$ value of 3.18 nM (Fig. 7c; Supplementary Fig. 12 and Table 8).

As mentioned above (Fig. 6b–d), Y98, A137, H141, A142, G143, and D190 residues use their main chain or side chain to form hydrogen bonds, and therefore, antibody binding was abolished upon their mutation. A144 mutated to Glu with a longer side chain affected its interaction with 12H5. Additionally, W153 makes hydrophobic interactions to stabilize its contact with the RBS. Taken together, we show that eight residues are strategic for C12H5 binding to the HA RBS, six of which (Y98, A137, H141, G143, W153, D190) are highly conserved (>90%), with two residues (A142, 77.4%; A144, 73.2%) moderately conserved. Combined, these residues have an average conservation of 90% across all H1N1 isolates (Supplementary Table 6).

To further investigate the structural basis of C12H5 H1 broad-neutralization and cross-neutralization to the H5N1 virus, we performed a sequence alignment and natural variation analysis for H5N1 strains (Supplementary Fig. 13; Supplementary Tables 7 and 9). The sequence analysis showed that HAs of H1N1 and H5N1 have some variable residues in the epitope region for C12H5. Specifically, we found five natural variant residues in the H5N1 strains, which may contribute to the subtype-specific binding of the H5N1 type: S133a, A137, Q142, R144, S145P (Supplementary Table 6). Thus, we also constructed these HA mutations in the QH/2005 strain and evaluated the resultant binding capacities using variations in $EC_{50}$ values. We found that the substitutions S133aL and S137A led to a slight decrease the binding reactivities for C12H5, with a nearly 2-fold increase in $EC_{50}$ values. The substitutions of Q142L, R144N, and S145P, however, completely abolished the binding to C12H5, with undetectable $EC_{50}$ values (Fig. 7d). These results suggest that Q142, R144, and S145 are key residues for C12H5 binding to the H5N1 virus. These three key sites (Q142, R144, and S145) are moderately conserved, about 67%, among the H5N1 isolates. The two equivalent positions in the H1 strains, A142 and A144, are also essential for C12H5 binding to H1N1, despite being different residues; another position, S145 might be a specific residue for H5N1 recognition.

In the 12H5: HAhr$^{CA}$ complex, we found that the R144N mutation could introduce N-glycosylation and result in steric occlusion with the light chain of 12H5 (Supplementary Fig. 14a). In addition, the Q142L mutation would change the electronic environment of the 140-loop (Supplementary Fig. 14b), and replacement of S145P would decrease the flexibility of the 140-loop and diminish its binding to C12H5 (Supplementary Fig. 14b). These three key sites are moderately conserved (67% among H5N1 isolates). Furthermore, based on the sequence identity of C12H5 for H1N1 and H5N1, nine (9/15) positions are preserved (i.e., the same amino acid types in both subtypes), suggesting that C12H5 may be able to tolerate high variations among the H1 and H5 strains. This likely also explains the cross-neutralization capacity of

C12H5 over other nAbs, such as 5J8. For example, at the 190 position—a key site associated with virus-host specificity—Asp is observed in ~91.6% of human H1 strains and Glu in ~99.7% of avian H5N1 strains. Yet, the known D190E mutation can abolish the binding of 5J8[20]; C12H5, however, can tolerate this polymorphism (Fig. 6f, h and Supplementary Fig. 9). We next carried out a mutagenesis analysis of the HA trimer by swapping the amino acid residue at position 190. The mutants, CA-D190E and QH-E190D, showed good reactivities against C12H5; albeit, with a 5- to 10-fold decrease in activity with respect to their corresponding wild-type HAs (Fig. 7e). Tolerance of the D/E190 polymorphism by C12H5 is also evident in the structural analysis (Supplementary Fig. 9), which illustrates the potential for the main chain of C12H5 G30c$^L$ to form favorable hydrogen bonds with the side chains of either H1-D190 or H5-E190.

We next sought to explore the relevance of the C12H5-like antibody in context of human immune responses to influenza. We measured the reactivities of human sera against the C12H5 epitope using a blocking ELISA. Five serum samples were taken from H1N1-infected persons who were H1N1-RNA positive; we were unable to obtain H5N1 human sera. Sera were pre-incubated with CA4/2009 or NC/1999 HA before addition of the enzyme-labeled antibody. We found that the blocking rates of human sera against C12H5 or FI6 were 39.2% and 40.2% for CA4/2009, the rates against C12H5 or CH65 were 18.8% and 30.4% for NC/1999 HA reactions, respectively (Fig. 7f). There is no significant difference between the C12H5 and the control antibodies, suggesting these bnAb-like human antibodies response upon the virus infections are relatively medium frequent.

The above-mentioned human sera were also used to probe the sensitivity of C12H5 bearing point mutations at critical sites. We found that six of the eight HA mutants (H141A, A142E, G143R, A144E, W153A, D190A) had significantly diminished reactivity to the 5 anti-H1 human sera (6- to 51-fold lower in the half-effective titers; $ET_{50}$). Of these, five mutations (A142E, G143R, A144E, W153A, D190A) are in critical sites in both H1 and H5 (Fig. 7g). Further, we show that human sera are not sensitive to two mutations (Y98A, A137E); albeit Y98 is part of the common epitope for H1 and H5 (Fig. 7g–i). Together with the aforementioned blocking assay, human antibodies in patients naturally infected with H1 may have a comparative affinity in targeting the broad neutralization epitope like C12H5, FI6 and CH65 epitopes. These results are consistent with the findings in the literature that broad neutralization epitopes seem to have relatively immunodominance in host immunity than strain-specific ones, and should be rationally designed for more immunogenic in broadly protection by means of immune focusing[11,12,33]. Overall, our selective mutagenesis assays revealed eight key residues for H1 binding (Y98, A137, H141, A142, G143, A144, W153, D190) (Fig. 7h), three critical sites for H5N1 binding (Q142, R144, S145) (Fig. 7i), and tolerance of the D/E residue at position 190 as key determinants for H1 and H5 co-neutralization; of these, six critical sites (H141, A142, G143, A144, W153, D190) are relevant to the human immune response.

## Discussion

Influenza viruses continue to be a threat to public health, and may add to the current health burden associated with the SARS-CoV-2 pandemic. SARS-CoV-2 and other deadly coronaviruses, SARS-CoV and MERS-CoV, are derived from zoonotic CoVs that have also crossed the species barrier[34]. This pandemic has better highlighted globally importance of zoonotic virus infection, which can be quite severe due to a lack of preexisting immunity in humans. Zoonotic influenza infections, such as H5N1[4] and H7N9[35], have caused several outbreaks over the past two decades. The antigenic evolution of influenza viruses poses a significant challenge for the development of highly effective and long-lasting vaccines or antiviral drugs. Broadly neutralizing antibodies that target conserved epitopes can serve as prophylactic or therapeutic reagents, and guide the design of universal vaccine

candidates with broad and long-lasting protection. Such a rationale will undoubtedly be useful for any new emergence.

In this work, we performed functional and structural studies using a chimeric antibody, C12H5, that targets the HA receptor-binding sites. We show that C12H5 offers broad-spectrum neutralizing reactivity against seasonal and pandemic H1N1, and offers partial cross-subtype neutralizing reactivity to H5N1 viruses both in vitro and in vivo. We compared its utility with the previously published broadly neutralizing antibodies that target the HA head of the H1 subtype, including CH65, 5J8, and Ab6649, and several antibodies (CR6261, F10, FI6) against both H1N1 and H5N1 strains that offer cross-type neutralization activity via targeting the HA stem region. Among these antibodies, we show the following: CH65 could neutralize seasonal influenza H1N1 viruses before 2009 in the testing virus panel; 5J8 could neutralizes the pandemic H1N1 but not some seasonal H1N1 strains; Ab6649, a broad acting H1 antibody, targets non-RBS conserved epitopes. As compared with these reported antibodies, C12H5 has broad neutralization breadth and an RBS-targeting nature similar to that of 5J8. However, the potency of C12H5 resembles that of CH65 for seasonal H1 strains, and 5J8 shows much higher HAI against the tested pandemic H1N1 strains than C12H5. Moreover, C12H5 uniquely co-recognizes the H5N1 strain to a better extent than do the tested H1 bnAbs 5J8, CH65 or Ab6649.

A recent study showed that a naturally occurring bnAb, FluA-20, has a conserved epitope at the interface between two adjacent head domains in the HA trimer, and functions to disrupt the integrity of the HA trimer through a neutralization mechanism[25]. The epitope is located at the 220-loop and the 90-loop, laterally adjacent to the RBS but not overlapping. Here, we found that the epitope of C12H5 overlaps with the 130-loop of the RBS domain, like 5J8, and at a site that is distant from that of the other H1 antibodies. However, unlike 5J8, the binding region of C12H5 is also uniquely located on the 140-loop outside the RBS pocket in addition to the 130-loop. In contrast, other antibodies recognize only the 130-loop. Despite the high level of variability in the 140-loop, this region may contribute to the cross-subtype reactivity of C12H5 via four key residues that are required for H5 co-recognition, including H/Y141 (90.5% conservation of H in H1, 99.4% of Y in H5) and G143 (99.5% in H1, 98.6% in H5). As such, the 140-loop may be an appealing target for the design of broad acting vaccines.

Through the co-crystal structure of 12H5:HAhr$^{CA}$, we were able to identify the structural basis for the broad H1 neutralization and cross-neutralization to H5 of 12H5, as well as the functional mimicry of receptor binding for virus infection inhibition. First, 12H5 principally interacts with HA, mostly at the 140-loop, through HCDR1, HCDR3, and LCDR1. In the interaction analysis, nine pairs of hydrogen bonds are distributed on the 140-loop of HA, and these bonds account for 60% of all hydrogen bonds in the interaction (9/15). Through mutagenesis analysis, we further showed that the continuous stretch H141-K145 in the epitope is critical for binding. This relatively linear stretch differs to that observed in other cross-subtype antibodies, and this may offer a starting point for antiviral molecule design. Second, 12H5 mimics the receptor-binding mode in a unique manner, forming three hydrogen bonds with the 130-loop via Tyr 32 in LCDR1; this is in stark contrast with the binding of the other receptor-mimicry antibodies, which use Asp and Glu carboxylate at the tip in HCDR2 or HCDR3, not the light chain[18–20,26,30,36,37] (Supplementary Fig. 15). This novel 12H5 receptor mimicry might offer an alternative solution for engineering a therapeutic antibody. Besides, the antigenic features in the receptor-binding region of HA between H1N1 and H5N1 are distinct due to functions in host recognition. E190D and G225E mutations are generally considered to be evolutionary sites for H1 viruses in their transition from recognizing avian receptors to recognizing human receptors[38]. Through our mutagenesis analysis, we show position 190 as the crucial site for 12H5 recognition. 12H5 can accept mutations at

residue 190 from aspartic acid to glutamic acid structurally by interacting with the main chain of G30c$^L$. Thus, these two amino acids do not collide with 12H5 in spatial conformations. Notably, the G30c$^L$ is an insertion in the antibody's mature affinity process (Supplementary Fig. 1). 12H5 can overcome antigenic variation among H1 and H5 subtypes and bear the mutations as related to receptor-binding transition from human to avian host. Other RBS-directed antibodies are not capable of recognizing this D190E mutation, and lose their neutralization ability. These implications suggest the potential of 12H5 as a structural template for subsequent small-molecule drug designs against human H1N1 and avian H5N1 viruses.

The current research on structural vaccinology is aimed at utilizing the conserved epitope of broadly neutralizing antibodies for the design of vaccine candidates that can stimulate a more robust and broader immunity in humans. The 12H5 epitope is highly conserved among most H1N1 and some H5N1 strains. The eight critical sites identified in the mutational and structural analyses are also highly conserved (mean, 90%). It is particularly challenging in this field to design universal vaccines against various subtypes of influenza viruses; indeed, the most advanced and promising methods are guided by the epitope structures identified by broadly nAbs[33,39,40]. In this study, C12H5 could broadly neutralize H1 and cross-neutralize H5 by engaging with a receptor-binding site that tolerates key site differences between H1 and H5. We suggest consideration of two key points for the design of an improved vaccine candidates. First, when selecting the recommended strains for traditional vaccine development during the influenza season, the antigenic composition of H1N1 strain should be prioritized to select the strain that harbors most of the C12H5 epitope residues; i.e., Y98, A137, H141, A/Q142, G143, A/R144, S145, W153, D/E190. Second, an immunogen designed for a broadly protective influenza vaccine—designed using an immune-focusing approach and based on structural information—could present the appropriate antigenic sites recognized by bnAbs such as C12H5. As suggested, based on its highly conserved and cross-neutralization nature, we sought to establish some clues that C12H5 epitope could be used in vaccine design. The C12H5 epitope features a conformational contour but its amino acid sequence is continuousness at two regions: R1, aa. K133a-V155 and R2, aa. H183-L194 (Supplementary Fig. 16a, b). We synthesized three polypeptides: two R1–R2 fusion peptides joined by a structurally favorable GSG linker, the sequences of which were derived from CA4/2009 and QH/2005 strains; and an R1 peptide of CA4/2009 strain. A fourth irrelevant polypeptide, VZV, served as a control. The polypeptides were then covalently conjugated to lysine residues of keyhole limpet hemocyanin (KLH) (Supplementary Fig. 16c), a widely used particle carrier for enhancing immunogenicity[41,42]. We then undertook immunization experiments: 10 mice were vaccinated with 25 μg of KLH-conjugated polypeptides at prime-boost immunization regimen delivered at weeks 0, 2, and 4. The sera obtained from immunization with CA-KLH, CA-R1-KLH, and QH-KLH conjugates showed 37% to 60% blocking ratios for C12H5 from binding to CA4/2009 HA (Supplementary Fig. 16d). Moreover, the antisera of QH-KLH exhibited high reactivity against QH/2005 HA (Supplementary Fig. 16e). Although these sera did not show observable neutralization titers in an in vitro neutralization assay (data not shown)—possibly due to a lower neutralizing antibody frequency —immunization with QH-KLH afforded 20% and 40% survival rates to vaccinated mice who were lethally challenged with MA-CA4/2009 and MA-QH/2005 viruses, respectively (Supplementary Fig. 16f–i). Additionally, immunization with CA-KLH or CA-R1-KLH provided 40% survival against lethal challenge with MA-CA4/2009 (Supplementary Fig. 16f). Taken together, these results suggest that the C12H5 epitope is otherwise immunogenic, and we believe

that a primary polypeptide constituted by partial C12H5 epitope could serve as a starting model for further rational design of H1/H5 broad vaccine candidates.

In conclusion, we elucidated an excellent H1 and H5 cross-neutralizing antibody that can mimic the human and avian receptor-binding modes. This antibody could serve as a template for further structure-guided drug discovery and design. The epitope binding regions may also help to guide the design of cross-subtype or universal influenza vaccines.

## Methods

### Viruses and cells
Biodefense and Emerging Infections Resourses Repository (BEI Resources) kindly provided A/Texas/36/1991 (H1N1), A/Beijing/262/1995 (H1N1), A/New Caledonia/20/1999 (H1N1), A/Solomon Is/3/2006 (H1N1), A/Brisbane/59/2007 (H1N1), A/California/04/2009 (H1N1), A/California/07/2009 (H1N1), A/Brisbane/10/2007 (H3N2) and B/Florida/4/2006 (Yamagata) strains. The University of Hong Kong kindly provided A/Hong Kong/134801/1994 (H1N1), A/Shantou/104/2005 (H1N1), A/Hong Kong/MB-1/2010 (H1N1), A/Hong Kong/402618/2011 (H1N1). The recombinant H5N1 viruses were constructed using frame of A/Puerto Rico/8/1934 (H1N1) containing HA and NA genes derived from A/Chicken/HK/YU22/2002 (H5N1), A/Bar-headed Goose/Qinghai/15C/2005 (H5N1), A/Migratory Duck/Jiangxi/2295/2005 (H5N1) or A/Xinjiang/1/2006 (H5N1). The First Affiliated Hospital of Xiamen University and the Xiamen International Travel Healthcare Center provided the A/Xiamen/N514/2009 (H1N1) and A/Xiamen/s27/2015 (H1N1) strains.

The mouse-adapted strains—MA-A/California/04/2009 (H1N1, MA-CA4/2009), MA-A/California/07/2009 (H1N1, MA-CA7/2009), MA-A/New Caledonia/20/1999 (H1N1, MA-NC/1999), and MA-A/Bar-headed Goose/Qinghai/15C/2005 (H5N1, MA-QH/2005) —were generated through successive passaging in the lungs of mice. All viruses were grown in MDCK cells using standard viral culturing techniques. Madin-Darby canine kidney (MDCK) cells were maintained in Eagle's minimal essential medium (MEM) supplemented with 10% calf serum. Cell lines used in this study were obtained from the ATCC (MDCK) or Thermo Fisher Scientific Inc. (CHO, 293 T, Sf9 and H5 cells). All cell lines used in this study were routinely tested for mycoplasma and found to be mycoplasma-free.

### Monoclonal antibodies (MAbs)
Seasonal H1N1 virus strains A/Hong Kong/134801/1994 (HK/1994), A/New Caledonia/20/1999 (NC/1999), and A/Brisbane/59/2007 (BR/2007) were chosen as immunogens for mAb production. Preparation of mAbs followed standard hybridoma technology, as previously described[22]. Briefly, 6-week-old female BALB/c mice were injected subcutaneously with formalin-inactivated HK/1994 and then, at twice weekly intervals, with successive equivalent doses of virus NC/1999. Fusion was performed two weeks after the final immunization with virus BR/2007. The resulting hybridomas were screened for the secretion of NC/1999-specific mAbs using a hemagglutination inhibition assay. We further screened for antibody cross-reactivity to other HA subtypes using a hemagglutination inhibition assay. The hybridoma cells were cloned using a limiting dilution at least three times, and positive clones were expanded and cultured in 75-cm² flasks. MAbs were prepared by injecting hybridoma cells into the peritoneal cavities of pristine-primed BALB/c mice; ascites were collected after 9–12 days and stored at −20 °C. One mAb, 12H5, neutralized the tested viruses in all of the assays. The hybridoma producing mAb 12H5 was cloned three times via limiting dilution, and 12H5 mAb was then purified from mouse ascites using protein A agarose columns (GE Healthcare).

### Sequencing analysis
Influenza RNA was extracted from samples using a QIAamp Viral RNA Mini Kit (Qiagen), following the manufacturer's instructions. The extracted viral RNA was subjected to one-step RT-PCR (QIAGEN One-Step RT-PCR Kit) with influenza A HA primer sets. The PCR products were separated on a 1.5% agarose gel with a 100-bp DNA ladder and visualized using a UV transilluminator. The PCR products were gel purified using a Universal DNA Purification Kit (TianGen) and sequenced. Sequencing of the variable gene regions of the 12H5 antibody was performed. Briefly, total RNA was extracted from 107 hybridoma cells using a MiniBEST Universal RNA Extraction Kit (TaKaRa Bio Inc.). The extracted RNA was subjected to a reverse transcription reaction with the following primers: 5′-GGGAATTCAT-GRAGWCACAKWCYCAGGTCTTT-3′ (L-chain-forward) 5′-CCCAAGCT-TACTGGATGGTGGGAAGATGGA-3′ (L-chain-reverse) for reverse transcription of the light chain variable region gene and 5′-GGGAATTCATGRASTTSKGGYTMARCTKGRTTT-3′ (H-chain-forward) 5′-CCCAAGCTTACGAGGGGGAAGACATTTGGGAA-3′ (H-chain-reverse) for reverse transcription of the heavy chain variable region gene.

### Construction of chimeric antibodies
Chimeric versions of 12H5 (C12H5), six previously reported influenza A HA antibodies (5J8, CH65, Ab6649, CR6261, FI6, and F10 antibodies), and a control antibody (C12G6) containing a human Fc fragment were constructed as described previously[23]. The variable gene for each antibody was inserted into a pTT5 vector containing the constant region of the human IgG1 gamma heavy chain or kappa light chain. Recombinant antibodies were expressed in Chinese hamster ovary (CHO) cells through transient transfection and purified from the culture media by MAbselect Sure (GE Healthcare) affinity chromatography.

### Hemagglutination inhibition (HAI) assay
Hemagglutination inhibition was performed under the World Health Organization Manual on Animal Influenza Diagnosis and Surveillance, modified as previously described[43,44]. Briefly, viruses were diluted to 8 HA units and combined with an equal volume of serially diluted mAbs and incubated for 1 h at room temperature. An equal volume of 0.5% Turkey red blood cells was added to the wells, and the incubation continued on a gentle rocking plate for 30 min. Button formation was scored as evidence of hemagglutination.

### Microneutralization assays
Cell-based microneutralization assays were performed as previously described[45]. Briefly, two-fold dilutions of mAbs were mixed with 100 $TCID_{50}$ of viruses and incubated for 1 h at room temperature. The mixture was added into a 96-well plate of confluent monolayers of MDCK cells. After 1 h adsorption, the virus inoculums were removed, and the cells were cultured with MEM containing trypsin (2 mg/mL). HA tests of the supernatant were scored as evidence of neutralization. For the HA test, 50 μL of 0.5% turkey red blood cells (TRBCs) was added to 50 μL of cell culture supernatant, and the mixture was incubated at room temperature for 1 h. The neutralization titers was determined as the lowest mAb concentration negative for hemagglutination. The assay was performed in quadruplicate.

### Direct and Sandwich ELISA
96-well plates were coated overnight at 4 °C with 2 μg/mL purified HAs (100 μL per well). The plates were washed three times with PBS containing 0.1% v/v Tween-20 (PBST) and blocked with 1× enzyme dilution buffer (PBS + 0.25% casein + 1% gelatin + 0.05% proclin-300) for 2 h at 37 °C. The plates were then washed with PBST. Wells were then incubated with serial 2-fold dilutions of purified antibody for 30 min at 37 °C. After three washes, 100 μL of horseradish peroxidase (HRP)-conjugated goat anti-mouse or anti-human IgG antibody (Abcam, 1:5000 dilution) solution was added to each well and incubated at 37 °C for 30 min. After five washes, 100 μL of tetramethylbenzidine (TMB) substrate (WANTAI BioPharm) was added at room temperature

in the dark. The reaction was stopped after 15 min with 2 M $H_2SO_4$ solution, and the absorbance was measured at 450 nm wavelength. All samples were run in triplicate. The $EC_{50}$ values were calculated with Prism software (GraphPad) using a non-linear regression analysis.

The binding profiles of HA and its mutants against MAb C12H5 were evaluated by double-antibody sandwich ELISA. The wells of a 96-well plate were coated with 200 ng of anti-His antibody (Proteintech, 1:5000 dilution) prepared in PBS, pH 7.4, and incubated overnight at 4 °C. HA proteins in half-serially diluted concentrations (starting from 5 µg/mL; diluted in PBS) were added to each well and incubated at 37 °C for 30 min. This was followed by the addition of horseradish peroxidase (HRP)-conjugated MAb C12H5 and a further 30-min incubation. The color was developed using TMB substrate, as described above. $EC_{50}$ values were calculated at the half-point of the curve by polynomial fitting.

### $K_D$ determination
$K_D$ values were determined by SPR technology using a Biacore3000 instrument (GE Healthcare), as described previously[26]. The anti-human IgG Fc antibody (GE Healthcare) was amine-coupled to a CM-5 sensor chip for use. MAb C12H5 was then captured on the sensor surface at a 30 µL/min flow rate in HBS buffer (10 mM HEPES, 150 mM NaCl, 3 mM EDTA, and 0.005% Tween-20, pH 7.4). The kinetics of C12H5 binding with HA and its mutants were measured at 30 µL/min flow rate in HBS buffer, delivered at 2-fold serially diluted concentrations (160, 80, 40, 20, and 10 nM). The flow durations were 200 s for the association stage and 10 min for dissociation. Association rates ($k_a$), dissociation rates ($k_d$), and affinity constants ($K_D$) were calculated using BIAcore evaluation software. All experiments were repeated twice. The mean $k_a$, $k_d$, and $K_D$ values are reported.

### Immunofluorescent staining
Monolayers of MDCK cells in 24-well plates were infected with HK/1994 (H1N1), A/California/04/2009 (H1N1, CA4/2009) or QH/2005 (H5N1) at a multiplicity of infection (MOI) of 0.2 for 16 h at 37 °C in the presence of 5% $CO_2$. Virus-infected cells were then fixed with 4% paraformaldehyde for 30 min in the dark, treated with 0.3% Triton X-100 diluted in PBS for 10 min, and then blocked for 1 h at 37 °C with 4% bovine serum albumin (BSA) in PBS. Cells were then incubated with C12H5, 5J8, or control antibody C12G6 for 1 h at 37 °C. After three rinses with PBS, the cells were sequentially stained with fluorescein isothiocyanate (FITC)-conjugated goat anti-human IgG antibody (KPL, 1:5000 dilution) for 30 min and 4, 6-diamidino-2-phenylindole (DAPI) for 5 min. Cells were then visualized using a fluorescence microscope using a Zeiss LSM 780 confocal microscope with the ZEN software (version 2.3).

### Prophylactic and the therapeutic efficacy studies in mice
In a prophylactic study, a group of 5 female BALB/c mice, aged 6−8 weeks, each received 10, 5, 2.5, or 1 mg/kg of C12H5 or 10 mg/kg of control IgG C12G6 (200 µL) one day before being intranasally challenged with 25 times the 50% mouse lethal dose ($MLD_{50}$) of MA-CA4/2009 (H1N1), MA-NC/1999 (H1N1) and MA-QH/2005 (H5N1) viruses delivered in a 50 µL volume. In therapeutic studies, C12H5 Mab, at a dose of 20, 10, 5 or 1 mg/kg, or 20 mg/kg of control IgG C12G6 was intravenously injected through the tail vein at 24 h after virus infection. Animals were observed daily for mortality and morbidity. Body weight was measured for up to 14 days after infection.

For a therapeutic comparison of the antibodies, C12H5 was injected at one of two doses 24 h after virus infection: 15 mg/kg or 3 mg/kg C12H5 to treat MA-CA4/2009 (H1N1), and 15 mg/kg C12H5 to treat MA-QH/2005 (H5N1). The Humane Endpoints guidelines were strictly followed for all in vivo experiments. Animals that lost more than 25% of their initial body weight were immediately euthanized by $CO_2$ asphyxiation and were recorded as nonsurvivors. The lungs of the

mice were collected for virus titration at 6 days after infection. 12H5 against H1N1 and H5N1 virus infection in a mouse model. For mouse antibody 12H5, female mice ($n = 5$) were challenged with lethal doses of MA-CA7/2009 (H1N1), and MA-QH/2005(H5N1)−and then treated with different doses (10 mg/kg, 5 mg/kg, or 2.5 mg/kg) of 12H5 or PBS (placebo) at 24 h after infection. Animals were observed daily for mortality, and body weight was measured for up to 14 days after infection.

### Selection of escape mutants
Seasonal H1N1 virus strains HK/1994, NC/1999, BR/2007, and CA7/2009 were selected to identify escape mutants. Escape mutants were selected by culturing the above virus strains in MDCK cells in the presence of MAb C12H5. Viruses were incubated with purified MAb C12H5 (final concentration of 10 mg/mL) for 1 h, with mixtures inoculated into confluent MDCK cells growing in 6-well tissue culture plates. After 1 h adsorption, the cells were overlaid with MEM containing 1% Bacto-Agar (Difco) and MAb C12H5 (final concentration of 50 µg/mL) and trypsin (2 mg/mL) and then incubated for 2 days at 37 °C. Escape mutants were purified from single isolated plaques and propagated in MDCK cells with serum-free MEM containing trypsin. The nucleotide sequences of the HA genes of the parent strains and the escape mutants were determined, and deduced amino acid sequences were compared among the viruses.

### Cloning, expression, and purification of the hemagglutinins
The genes encoding for the ectodomains of HA proteins (aa 11-329 of HA1 and 1-174 of HA2, H3 numbering) from the CA4/2009 (H1N1), NC/1999 (H1N1), A/Beijing/262/1995 (H1N1, BJ/1999), A/Washington/05/2011 (H1N1, WA/2011) and QH/2005 (H5N1) were cloned into the baculovirus transfer vector (pAcGP67B) (BD Biosciences) for baculovirus preparation. The construct contains an N-terminal gp67 secretion signal peptide and a C-terminal thrombin, trimerization Foldon sequence, and 6-His-tag for purification, as previously described[26]. For the expression of recombinant HA proteins, H5 suspension cells (Invitrogen) were infected with recombinant baculovirus at 5−8 MOI at 28 °C for 72 h. The supernatant was then dialyzed against PBS, pH 7.4, and purified with Ni-NTA resin (GE Healthcare) with 250 mM imidazole elution. For immune complex preparation, purified BJ/1995 HA was treated with thrombin to remove the foldon before trypsin digestion, as previously described[26]. CA4/2009 HA was prepared without protease treatment to maintain the trimer formation.

### Purification, crystallization, and structure determination
The 12H5 Fab was prepared by papain digestion of mAb 12H5 and purified with DEAE-5PW (TOSOH Biosciences), as described previously[46]. HA was mixed with 12H5 Fab in a molar ratio of 1:1.2 (HA protomer to Fab) and incubated at 37 °C for 2 h. The immune complex was further purified to remove excess Fab by gel filtration on a Superdex 200 column (GE Healthcare) in 10 mM Tris, pH 8.0, with 50 mM NaCl. The complex was concentrated to ~7.5 mg/mL for crystallization. Crystallization was performed using sitting-drop vapor diffusion in the screening stage and hanging drop in microseeding optimization at 20 °C. Crystals of the complex were grown in 0.1 M MES, pH 6.5, with 13% (w/v) PEG 2000. The crystal growth of the immune complex took about 2 months for final data collection. Crystals were cryo-protected in reservoir solution supplemented with 30% glycerol at 100 K before collecting the diffraction data. Diffraction data were collected at the Shanghai Synchrotron Radiation Facility (SSRF) beamline BL17U using a Quantum-315r CCD Area Detector. Datasets were processed using the HKL-2000 program package (http://www.hkl-xray.com). The unit cell of 12H5:HAhr$^{CA}$ crystal can only accommodate one HA head region and one 12H5 Fab in its asymmetric unit. During molecular replacement phasing in Phenix[47], we found that the

ensemble comprised one HA head region and one Fab molecule, and we could not trace the rest of the HA monomer during model building in Coot[48]. We surmise that the long-term crystallization process of ~2 months may have led to the degradation of the HA trimer, as also observed in our previous study[26]. We used sodium dodecyl sulfate-polyacrylamide gel electrophoresis (SDS-PAGE) to show HA degradation. We characterized the crystallization sample of the 12H5:HA$^{CA}$ complex before crystallization and at 2 months after storage (Supplementary Fig. 7e). We identified a band in the SDS-PAGE at a molecular weight of ~70 kD (corresponding to one Fab 47kD + one HA head region 23 kD, spanning aa 56–263 of HA1 in the final model) in the asymmetric unit. We used this permutated complex of partial HA and one Fab for phase searching with molecular replacement (MR), implemented with PHASER[49] suite using PHENIX[47] to refine the initial phases. The search models for the HA head region and 12H5 Fab were PDB code: 3lzg and 1c1e, respectively. The resulting models were manually built in Coot, refined with PHENIX and analyzed with MolProbity[50]. In brief, one round of rigid-body refinement was performed after MR. The refined models were manually modified in Coot; coordinates and individual B factors were refined in reciprocal space. TLS refinement was performed in the later stages with auto-searched TLS groups in PHENIX, which are listed in REMARK 3 sections in the deposited PDB files. Data collection and structure refinement statistics are summarized in Supplementary Table 3.

## Cryo-EM and 3D reconstruction

The 12H5: HA complexes were prepared as described above. The cryo-EM and 3D reconstruction were performed as per our published studies[51]. In brief, an aliquot (3-μL) of complex sample was deposited onto a glow-discharged Quantifoil holey carbon grid (R2/1, 200 mesh; Quantifoil Micro Tools). After 6 s of blotting to remove the excess sample, the grid was plunge-frozen into liquid ethane using a Thermo Fisher Vitrobot, and then examined under low-dose conditions at 300 kV with an Thermo Fisher F30 transmission electron microscope. Images were recorded on a Gatan K3 direct electron detector (36-frame movie mode) at a nominal magnification of 39,000 at supre-resolution mode corresponding to a pixel size of 0.389 Å. A total electron dose of ~60 e⁻/Å² was used, with an exposure time of 4.5 s. SerialEM[52] automated data collection software was used for all data acquisition.

Motion correction, CTF estimation, particle picking and extraction were performed with CyroSPARC V3[53]. Micrographs with excessive drift or astigmatism were discarded. Two rounds of reference-free 2D classification, initial model and final 3D density map reconstruction was all performed with CyroSPARC V3. The final map resolution was determined based on "gold-standard" criteria of the Fourier Shell Correlation (FSC) curve with a cut-off at 0.143[54]. Local map resolution was estimated with ResMap[55].

## Atomic model building, refinement, and 3D visualization

The crystal of 12H5: HAhr$^{CA}$ and CA trimer (PDB code: 3LZG) were used to generate an initial model by homology modeling unsing Accelrys Discovery Studio software (available from: URL: https://www.3dsbiovia.com). We initially fitted the initial model into the corresponding final cryo-EM map using Chimera[56], and further corrected and adjusted them manually by real-space refinement in Coot[48]. The resulting models were then refined with phenix.real_space_refine in PHENIX[47]. These operations were executed iteratively until the problematic regions, Ramachandran outliers, and poor rotamers were either eliminated or moved to favored regions. The final atomic models were validated with Molprobity[50]. All cryo-EM relative figures were generated with Chimera or ChimeraX[57].

## Sequence analysis and epitope conservation

A total of 8907 and 2263 full-length, non-redundant H1N1 and H5N1 HA sequences, respectively, were downloaded from the NCBI FLU database on 3rd May 2021. After multiple sequence alignment, a residue was considered conserved only if it was identical at the equivalent site among most sequences. All sequences were aligned and analyzed by Clustal Omega. The values reported for percent conservation are the number of sequences with an identical change at a position divided by the total number of sequences. The conservation figure indicating the sequence conservation was generated using PyMoL Molecular Graphics System.

## Gel filtration chromatography

TSK Gel PW5000xl 7.8×300-mm columns (TOSOH Corporation) were equilibrated in PBS using the Waters HPLC system. Samples of HA, Fab and the immune complexes were dialyzed against PBS and diluted to 1 mg/mL before loading. The flow rate was maintained at 0.5 mL/min and the absorbance at 280 nm was recorded.

## Sodium dodecyl sulfate-polyacrylamide gel electrophoresis (SDS-PAGE)

Protein samples were mixed with loading buffer (50 mM Tris pH 6.8, 2% SDS, 5% 2-mercaptoethanol, 0.01% bromophenol blue, 8% glycerol), boiled for 10 min, and subjected to SDS-PAGE. Equal amounts of protein for each sample were loaded onto SDS-PAGE gels. The proteins were electrophoresed for 70 min at 120 V in a BioRad MINI-PROTEAN Tetra system (BioRad Laboratories), and the gel was stained with Coomassie Brilliant Blue R-250 (Bio-Rad) for 30 min at room temperature.

## Cell-based entry inhibition assay

The cell-based entry inhibition assay was performed as described previously[23]. Briefly, viruses were incubated for 1 h with 10 μg/mL C12H5 or polyclonal rabbit serum diluted in PBS raised against purified NC/1999, HK/1994, CA4/2009, A/XM/N514/2009 (H1N1, XM/2009), A/Hong Kong/MB-1/2010 (H1N1, HK/2010), QH/2005 or A/Xinjiang/1/2006 (H5N1, XJ/2006) viruses. The mixture was added to MDCK cell monolayers in 96-well format in infection medium (DMEM supplemented with 2 μg/mL acetylated trypsin). The inoculum was subsequently removed and replaced with antibody or polyclonal rabbit serum at the indicated concentrations, and cultured for 16–18 h at 37 °C in 10% CO$_2$. Supernatants were then removed, and the cells were fixed in 80% acetone and stained sequentially with a mouse monoclonal anti-NP primary antibody and Alexa Fluor 488-coupled anti-mouse secondary antibody (Invitrogen, 1:5000 dilution). Cellular nuclei were then labeled with DAPI, and the plates were analyzed using an Opera Phenix High-Content Screening System (PerkinElmer).

## Egress inhibition assay

Egress inhibition assay was carried out as described previously[25]. Briefly, monolayers of MDCK cells were seeded into 96-well plates in DMEM supplemented with 10% calf serum for 12 h at 37 °C. The cells were replenished with infection medium (DMEM supplemented with 2 μg/mL acetylated trypsin) and infected with viruses at an MOI of 2 for NC/1999, HK/1994, CA4/2009, XM/2009, HK/2010, QH/2005 or XJ/2006 viruses for 4 h at 37 °C in 10% CO$_2$. The supernatants were then removed and the cells were washed three times with PBS to remove non-internalized virus particles. The cells were replenished with an infection medium containing serial dilutions of C12H5 or a control antibody. After 18 h at 37 °C in 5% CO$_2$, the supernatants were collected and clarified at 300 ×$g$ for 15 min to remove the cell debris. Serial 2-fold dilutions of supernatants in PBS were added to an equal volume of 0.5% turkey red blood cells for 30 min to determine the virus titres by Hemagglutination assay. Button formation was scored as evidence of hemagglutination.

## Polypeptide-carrier protein conjugate preparation

The C12H5 epitope features a conformational contour but its amino acid sequence is continuousness at two regions: R1, aa. K133a-V155 and R2, aa. H183-L194 We synthesized three polypeptides: two R1–R2 fusion peptides joined by a structurally favorable GSG linker, the sequences of which were derived from CA4/2009 and QH/2005 strains; and an R1 peptide of CA4/2009 strain. A fourth irrelevant polypeptide, VZV, served as a control. The polypeptides were then covalently conjugated to lysine residues of keyhole limpet hemocyanin (KLH) To prepare polypeptide-carrier protein conjugates, all the polypeptide were conjugated to a lysine residue on the carrier protein keyhole limpet hemocyanin (KLH) using m-maleimidobenzoyl-N-hydroxysuccinimide ester (MBS), following the manufacturer's protocol.

## Immunization assay

Six groups of BALB/c mice ($n = 10$ per group) around 7–8 weeks old were immunized in 2-week intervals with KLH-conjugated C12H5 epitope peptides, which were formulated with Freund's adjuvant (Sigma-Aldrich). A dosage of 25 µg peptides per 200 µL was administered by intraperitoneal (IP) route for all these animals. Sera samples were collected before each injection and stored for ELISA and other analyses. At week 5, every one group was divided into two subgroups ($n = 5$ per group) randomly, and each subgroup was then intranasally challenged with 25 times amount of the 50% mouse lethal dose (MLD50) of MA-CA4/2009 (H1N1) or MA-QH/2005 (H5N1) viruses, which were prepared in a 50 µL volume. The mice were observed daily for mortality and morbidity, and body weight of survival animals was measured for up to 14 days after infection, as described above.

## Blocking ELISA

The reactivities of human or mice sera against the C12H5 epitope were measured by a blocking ELISA. Five serum samples were taken from H1N1-infected persons who were H1N1-RNA positive; we were unable to obtain H5N1 human sera. Sera were pre-incubated with H1 HA before addition of the antibody. The sera harvested from the mice immunized with C12H5 epitope peptides were measured by the blocking ELISA assay as well. In brief, the concentration of the blocked enzyme-labeled antibody (C12H5-HRP, FI6-HRP or CH65-HRP) was first modulated to have an OD value between 0.8 and 1.2 without any blocking. Then, the human or mice sera (at 1:50 dilution) were coated to the wells with purified CA4/2009 or NC/1999 HA in blocking solution (final volume: 100 µL per well), and the microplates were incubated at 37 °C for 30 min. HRP-conjugated antibody was then added and incubated at 37 °C for 30 min. Wells were washed twice and the reaction was catalyzed with o-phenylenediamine substrate at 37 °C for 10 min. The OD value was converted to percentage inhibition using the following formula: PI (%) = 100 − [(OD$_{sample}$/OD$_{control}$) × 100]%, where the OD values of control wells were measured for the wells containing only HRP-conjugated antibody.

## Ethic statement

All infectious materials were handled in a BSL-2 facility under approved protocols according to Xiamen University guidelines. The participants providing serum samples were recruited from The First Affiliated Hospital of Xiamen University, China. Five serum samples were taken from H1N1-infected persons who were H1N1-RNA positive. The use of human sera in this study was approved by the Research Ethics Committee of Xiamen University (Approval NO. XMULAC20200232), all participants provided written informed consent.The experimental protocols were approved by the Xiamen University Laboratory Animal Management Ethics Committee. All manipulations were strictly conducted in compliance with the animal ethics guidelines and approved protocols. Mice were kept on a 12-h light/12-h dark cycle, at 22–24 °C and 30–70% humidity, with ad libitum access to food and water.

## Reporting summary

Further information on research design is available in the Nature Research Reporting Summary linked to this article.

## Data availability

The X-ray structure generated in this study have been deposited in the Protein Data Bank (PDB) database under accession code 7FAH for the 12H5: HAhr$^{CA}$ complex, the EM map of the 12H5: HA$^{BJ}$ and coordinate has been deposited in the Electron Microscopy Data Bank (EMDB) under accession code EMD-33831 and the PDB database for 12H5: HA$^{BJ}$ under accession code 7YHK. The experiment data generated in this study are provided in the Supplementary Information and Source Data file. Other publicly available data: 3LZG; 4K63; 3UBE; 4M5Z; 5UGY; 4HKX; 5W6G; 6A67; 5DUP; 4KTH; 5E30; 4K62; 3SM5; 2VIR; 1KEN; 4O5I. Source data are provided with this paper.

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

## Acknowledgements

This work was supported by grants from the National Key Research and Development Program of China (grant no: 2021YFC2301404), the National Natural Science Foundation of China (grant nos. 81871651; 82041038; 32000649). We thank the members in beamline BL17U1 at the Shanghai Synchrotron Radiation Facility) for the assistance in X-ray data collection.

## Author contributions

Y.G., Y.C, S.L., and N.X. designed the study. T.L., J.C., Q.Z., W.X., L.Z., R.R., S.Z., Q.W., Minqing H., Maozhou H., Y.Z., Z.L., Z.Z., X.C., J.L., Y.H., H.W., J.T., D.Y., L.C., and L.Z. performed experiments. T.L., J.C., Q.Z., W.H., H.Y., Y.G., Y.C., S.L., and N.X. analyzed data. T.L., Y.G., Y.C., and S.L. wrote the manuscript. T.L., J.C., Q.Z., W.X., L.Z., Y.W., J.Z., Y.G., Y.C., S.L., and N.X. participated in discussion and interpretation of the results. All authors contributed to experimental design.

## Competing interests

The authors declare no competing interests.
