## [Peer Review File · Nature Communications]

Reviewer comments, first round review

Reviewer #1 (Remarks to the Author):

In this manuscript, Li et al. report interesting functional and structural studies on a mouse monoclonal antibody, 12H5, elicited from three seasonal H1N1 virus strains HK/1994, NC/1999 and BR/2007. A mouse/human chimeric form of this antibody, C12H5, was successfully made from variable region of 12H5 and human IgG1 Fc region for functional analysis. The research work is extensive and systematic using varieties of in vitro and in vivo research tools, and the results are solid.

12H5 binds to influenza hemagglutinin head receptor binding site (RBS). C12H5 neutralizes and protects mouse from seasonal and pandemic H1N1 viruses and some H5N1 viruses. To compare with documented best H1N1 RBS antibodies 5J8 and CH65, which are pan-H1N1 human antibodies, c12H5 demonstrated overall better breadth for H1N1 strains, as well as H5N1 strains to which these other two human antibodies are not responsive. However, the neutralization potency of C12H5 is not always better, for example 5J8 does better in some assays, for example in HAI against pandemic H1N1 strains.

The neutralization mechanism is proposed to be hemagglutinin inhibition and virus egress. From X-ray and EM structural studies, 12H5 binds to the H1 RBS including the 140-loop, which is not involved in binding with 5J8 and CH65. 12H5 mimics the receptor binding mode with Tyr32 in LCDR1 which is unique as other RBS antibodies use Asp or Glu from HCDR2 or HCDR3. The structural and mutational analysis gives some clues as to the basis of C12H5's extended breadth to H5N1 viruses.

Although 12H5 is a mouse antibody, this study advances antibody breadth over individual best head binding antibodies to H1N1 strains. In the discussion, some polypeptides based on 12H5 epitope were tested and found to be immunogenic and could be used as starting points for further vaccine design.

Some minor points that should be addressed:

Page 2, line 40. Please replace "benefit for" with "could benefit".

Page 3, line 50. Please delete "strain".

Page 3, line 51. Please replace "strains" with "types (e.g. influenza B)".

Page 7, lines 151-152. "The DNA sequences of the VL (variable region of immunoglobulin light chain) and VH (heavy chain) of 12H5 were then acquired" So these were not known in the above analysis on mAb 12H5?

Page 8, lines 174-175. "Most notably, C12H5 showed heterosubtypic binding activity to H5 QH/2005 HA with a KD of 127 nM (Fig. 1B; Table S2), indicating a moderate interaction". However, the off rate is quite fast that could restrict its effectiveness and potency in some assays as well as in vivo.

Page 9, lines 190-191. "Of all the antibodies, CH65 displayed the most robust HAI activity but only to the seasonal H1 strains". 5J8 also showed just as robust activities to the pandemic H1N1 strains and CH65 and 5J8 both did better against many viruses compared to C12H5 within these subgroups.

Page 9, lines 194-197, "In contrast, our chimeric C12H5, however, showed high HAI activity against all major H1N1 viruses,.. " but from Fig. 1C, it did not show high HAI against SI/2006 and BR/2007 using the current high/moderate HAI definition.

Page 10, line 208. "suggesting that C12H5 is more potent than CL6649 and CR6261" At least in

this assay.

Page 10, lines 212-213. "Moreover, C12H5 exclusively offered cross-neutralization against the H5N1 virus; none of other H1-region antibodies recognized H5". This statement is misleading as CR6261, F10 and FI5 all neutralized H5 strains.

What is meant by none of the other H1-region antibodies? I assume H1 head-region antibodies that might address the above comment.

Page 10, lines 216-217. "and has overall a higher neutralization potency than the three stem bnAbs tested". Yes, true for overall but misleading as 5J8 and CH64 outperform C12H15 in individual cases. To my mind, overstating the results unnecessarily diminishes the conclusions.

Page 11, lines 240-241. "For the H5 virus, doses of 10 mg/kg and above effectively alleviated the weight loss observed". Any comment on why the body weight loss is shifted by several days compared for example to NewCal99 in Fig. 2D?

Page 12, lines 247-248. "therapeutic experiments collectively indicated that C12H5 could protect mice against H1N1 and H5N1 viruses in vivo". But much less effectively to H5.

Page 12, line 260. Please replace "loss" with "loss."

Page 13, lines 265-268. "Unexpectedly, mice treated with 5J8, while also presenting with low virus titers in the lung following MA-CA4/2009 (pdm H1N1) challenge, were not fully protected from H1 lethal challenge, recording only a 60% survival rate". Was this repeated due to its unexpected nature and inconsistency with other data reported here.

Page 13, lines 283-284. "C12H5 at a concentration of 0.1 mg/mL inhibits the egress of all above-mentioned viruses from the infected cells". Again, seems to be an overstatement of the data. There appears to be no egress inhibition of FL/2004.

Page 14, lines 290-291. "Furthermore, C12H5 can effectively block virus egress, as tested with five H1 and two H5 strains. Seems to be misleading- C12H5 doesn't block FL/2004 egress.

Page 14, lines 305-306, RMSD of 0.526 Å. Please round to 0.5 Å as three decimal points are not meaningful.

Page 15, lines 313-314. "12H5 indeed targets the RBS of different HAs". Do you mean different protomers within the same HA rather than different HAs?

Page 15, line 313. Please replace "cyro-EM" with "cryo-EM"

Page 15, line 318. Please replace "Table. S6" with "Table S6".

Page 16, line 339. Please replace "salt-bridging" with "salt bridges".

Page 16, line 341. Please replace "bonding" with "bond".

Page 16, line 342. Please replace "bonding" with "bond".

Page 16, line 343. Please replace "binding" with "interactions".

Page 17, line 353. Please replace "sites on" with "site".

Page 17, line 354. "interaction mimics the interaction of Sia-1". There is some mimicry is H bonds but I would say 'mimics to some extent'.

Page 17, line 356. Please replace "GlaNac" with "GalNac"?

Page 17, line 357. Please replace "meditates" with "mediates".

Page 19, line 402. "strong" I don't think there are any data on strength.

Page 19, line 404. Please replace "mutates" with "mutated".

Page 19, lines 409-410. "Combined, these residues have an average conservation of 90% across all H1N1 isolates". Does this include the 1918 and other early isolates. With this high degree of conservation, does the antibody not bind 1918 and other early H1N1 viruses?

Page 21, line 443. Please replace "a known" with "the known".

Page 22, line 477. Please replace "seem relatively" with "seem to have relatively".

Page 22, lines 478-480. "reciprocal to strain-specific neutralizing antibodies must be redesigned by means of immune focusing during broadly protective immunogen design^{11,12,31}". Please rephrase this whole sentence.

Page 24, lines 510-511. "CH65 could only neutralize seasonal influenza H1N1 viruses before 2009". What about 1918 and other H1N1 arising from that pandemic?

Page 25, line 535. Please insert `,' before we

Page 26, line 557. Please delete "190". Please replace "mutations from" with "mutations at residue 190 from".

Page 26, lines 561-562. "and bear the mutations as related to receptor binding transition". ????

Page 27, line 581. Please replace "stain" with "strain".

Page 37, line 800. Please replace "murmuring" with "numbering".

Page 38, lines 816-817. "HA was mixed with 12H5 Fab in a molar ratio of 1:1.2". Is this the molar ratio to HA protomer and not to HA trimer?

Page 39, line 837. Please delete "JV".

Page 39, line 844. Please replace "Phenixto" with "Phenix to".

Page 48, line 1056, Reference 24. Please abbreviate journal reference.

Page 57, line 1197. Fig. 5 title. "Structure of 12H5:HAhrCA complex" could be changed to "Structures of 12H5:H1 HA complexes", as the section is about crystal and EM studies of 12H5 to two different HAs: HAhrCA and BJ/1995.

Page 60, Fig. 7, line 1247. Please replace "Tale" with "Table".

Supplementary Materials

Page 25, Table S6. Please truncate to no more than one decimal point for the BSA.

Reviewer #2 (Remarks to the Author):

In the present study, Li and colleagues, isolated and characterized a murine antibody 12H5 that cross neutralizes H1N1 and H5N1 viruses, both belonging to group 1 influenza viruses. The authors showed that the mAb engages the receptor binding site (RBS) of both H1 and H5 hemagglutinins and that it is insensitive to the Asp->Glu amino-acid substitution at position 190 which explains, in part, the broad cross-reactivity of this mAb.

While this is a detailed characterization of an interesting antibody, it is simply that – an interesting mouse antibody which seems to have been serendipitously isolated (sequential immunization with seasonal H1N1 viruses then simply screening of a lot of hybridoma clones). The study does not teach us anything new about the human immune system, nor about strategies of how to induce this class of antibodies in animal and in humans.

Major:

The X-ray data processing and refinement statistics are inconsistent in the 2 tables provided. For example, the PDB deposition report shows R_w/R_f of 26.6/28.9 and the table S1 of the MS 26.9/29.2. Other inconsistencies like data resolution range are also noted.

It is surprising that the cryo-EM reconstruction was achieved to only 7Å of nominal resolution with over 300k particles and pixel size 1.12Å. The methods are inadequately described, and it is therefore impossible for me to judge what might have been the limiting factor in the cryo-EM data processing. I dare say that with more rigorous data processing (for ex. trying C3 symmetry, 3D classification and focused refinement these data should yield a 4Å reconstruction (or better). The MS is also lacking the data processing pipeline, estimates of local resolution and model to volume fit figures. Additionally, cryoSPARC is misspelled in the methods.

A more detailed structural comparison with 5J8 and CH67 (H1 specific) and FLD21.140 and AVFluIgG03 (H5 specific) is warranted.

Minor:

Why are Supplementary figures called out first? It would be more informative to re-work the main Figure 1 and include the protection data. SPR data can be summarized in a table.

CL6649 is the name of the clonal lineage, the mAb used is Ab6649.

Response to Reviewer Comments on the manuscript [NCOMMS-22-11224-T]:

We thank the two reviewers for recognizing the merit of our work and for their suggestions to improve our manuscript. We have fully addressed the comments with appropriate additional experiments and analyses. To facilitate the navigation of this document, we have copied the reviewers' comments verbatim in blue and typed our responses in **black**, and some figures related to their corresponding comments are copied here as well.

Reviewer 1

In this manuscript, Li et al. report interesting functional and structural studies on a mouse monoclonal antibody, 12H5, elicited from three seasonal H1N1 virus strains HK/1994, NC/1999 and BR/2007. A mouse/human chimeric form of this antibody, C12H5, was successfully made from variable region of 12H5 and human IgG1 Fc region for functional analysis. The research work is extensive and systematic using varieties of in vitro and in vivo research tools, and the results are solid.

12H5 binds to influenza hemagglutinin head receptor binding site (RBS). C12H5 neutralizes and protects mouse from seasonal and pandemic H1N1 viruses and some H5N1 viruses. To compare with documented best H1N1 RBS antibodies 5J8 and CH65, which are pan-H1N1 human antibodies, c12H5 demonstrated overall better breadth for H1N1 strains, as well as H5N1 strains to which these other two human antibodies are not responsive. However, the neutralization potency of C12H5 is not always better, for example 5J8 does better in some assays, for example in HAI against pandemic H1N1 strains.

The neutralization mechanism is proposed to be hemagglutinin inhibition and virus egress. From X-ray and EM structural studies, 12H5 binds to the H1 RBS including the 140-loop, which is not involved in binding with 5J8 and CH65. 12H5 mimics the receptor binding mode with Tyr32 in LCDR1 which is unique as other RBS antibodies use Asp or Glu from HCDR2 or HCDR3. The structural and mutational analysis gives some clues as to the basis of C12H5's extended breath to H5N1 viruses.

Although 12H5 is a mouse antibody, this study advances antibody breadth over individual best head binding antibodies to H1N1 strains. In the discussion, some polypeptides based on 12H5 epitope were tested and found to be immunogenic and could be used as starting points for further vaccine design.

Response: We thank the reviewer for the encouraging comments on our work.

Some minor points that should be addressed:

Comment 1: Page 2, line 40. Please replace "benefit for" with "could benefit".

Response: Corrected. (Page 2, line 40)

Comment 2: Page 3, line 50. Please delete "strain".

Response: Corrected. (Page 3, line 50)

Comment 3: Page 3, line 51. Please replace "strains" with "types (e.g. influenza B)".

Response: Corrected. (Page 3, line 51)

Comment 4: Page 7, lines 151-152. "The DNA sequences of the VL (variable region of immunoglobulin light chain) and VH (heavy chain) of 12H5 were then acquired" So these were not known in the above analysis on mAb 12H5?

Response: We usually sequence the DNA sequence of an antibody after a comprehensive characterization. These analyses don't require for sequence information. Thus, the DNA sequencing and other analyses are separated. To avoid any confusion on the unnecessary time logic, we have removed

the time conjunction word “then” in the sentence, now reads: “The DNA sequences of the VL (variable region of immunoglobulin light chain) and VH (heavy chain) of 12H5 were acquired and aligned with the most adjacent germline sequence using the IMGT database (<http://www.imgt.org>).” (Page 7, line 151)

Comment 5: Page 8, lines 174-175. “Most notably, C12H5 showed heterosubtypic binding activity to H5 QH/2005 HA with a KD of 127 nM (Fig. 1B; Table S2), indicating a moderate interaction”. However, the off rate is quite fast that could restrict its effectiveness and potency in some assays as well as in vivo.

Response: As suggested, we have revised the sentence as: “Most notably, C12H5 showed heterosubtypic binding activity to H5 QH/2005 HA with a KD of 127 nM (Fig. 1B; Table S2), indicating a moderate affinity within the range that has been reported to effectively neutralize the virus (KD < 250 nM) (Fig. 1f), while its relatively fast off rate might restrict its effectiveness and potency in some assays.” (Page 8, line 175)

Comment 6: Page 9, lines 190-191. “Of all the antibodies, CH65 displayed the most robust HAI activity but only to the seasonal H1 strains”. 5J8 also showed just as robust activities to the pandemic H1N1 strains and CH65 and 5J8 both did better against many viruses compared to C12H5 within these subgroups.

Response: As suggested, we have revised the sentence as: “Of all the antibodies, CH65 displayed the most robust HAI activity but only to the seasonal H1 strains. 5J8 also showed just as robust activities to the pdm H1N1 strains, CH65 and 5J8 both did better against many viruses compared to C12H5 within these subgroups.” (Page 9, line 190)

Comment 7: Page 9, lines 194-197, “In contrast, our chimeric C12H5, however, showed high HAI activity against all major H1N1 viruses,..” but from Fig. 1C, it did not show high HAI against SI/2006 and BR/2007 using the current high/moderate HAI definition.

Response: As suggested, we have revised the sentence as: “In contrast, our chimeric C12H5, however, showed high HAI activity against all major H1N1 viruses except for SI/2006 and BR/2007”. (Page 9, line 197)

Comment 8: Page 10, line 208. “suggesting that C12H5 is more potent than CL6649 and CR6261” At least in this assay.

Response: As suggested, we have revised the sentence as: “suggesting that C12H5 is more potent than Ab6649 and CR6261 at least in this assay.” (Page 10, line 209)

Comment 9: Page 10, lines 212-213. “Moreover, C12H5 exclusively offered cross-neutralization against the H5N1 virus; none of other H1-region antibodies recognized H5”. This statement is misleading as CR6261, F10 and F15 all neutralized H5 strains.

What is meant by none of the other H1-region antibodies? I assume H1 head-region antibodies that might address the above comment.

Response: Sorry for the typo. Yes, they are other H1 head region antibodies. We have revised the sentence as: “Moreover, C12H5 exclusively offered cross-neutralization against the H5N1 virus; none of other H1 head-region antibodies recognized H5.” (Page 10, line 212)

Comment 10: Page 10, lines 216-217. “and has overall a higher neutralization potency than the three stem bnAbs tested”. Yes, true for overall but misleading as 5J8 and CH64 outperform C12H5 in individual cases. To my mind, overstating the results unnecessarily diminishes the conclusions.

Response: We have toned down the sentence as “Taken together, despite the varied neutralizing potency among the different strains, we show that C12H5 outperforms the other three reported H1 head-region bnAbs in terms of neutralization breadth and has an overall higher neutralization potency than the three stem bnAbs tested (CR6261, F10, FI6). (Page 10, line 214)

Comment 11: Page 11, lines 240-241. "For the H5 virus, doses of 10 mg/kg and above effectively alleviated the weight loss observed". Any comment on why the body weight loss is shifted by several days compared for example to NewCal99 in Fig. 2D?

Response: We have added the comment in manuscript: “The body weight loss is shifted by several days compared to that MA-NC/1999, which might be due to resulting pathogenicity varying even at the same dose of antibodies across different virus strains.” (Page 12, line 243)

Comment 12: Page 12, lines 247-248. "therapeutic experiments collectively indicated that C12H5 could protect mice against H1N1 and H5N1 viruses in vivo". But much less effectively to H5.

Response: We have revised the sentence as: “The prophylactic and therapeutic experiments collectively indicated that C12H5 could protect mice against H1N1 and H5N1 viruses in vivo, despite less effectively to H5.” (Page 12, line 249)

Comment 13: Page 12, line 260. Please replace "loss" with "loss."

Response: Corrected. (Page 12, line 264)

Comment 14: Page 13, lines 265-268. "Unexpectedly, mice treated with 5J8, while also presenting with low virus titers in the lung following MA-CA4/2009 (pdm H1N1) challenge, were not fully protected from H1 lethal challenge, recording only a 60% survival rate". Was this repeated due to its unexpected nature and inconsistency with other data reported here.

Response: Thank you for pointing this out. We have not repeated this assay, as the results sounded normal in reasonable number of animals (N=5) similar with other antibodies. Actually, there is no the relatedness data for the virus titer vs. protection efficacy by 5J8 in the literature, Krause and colleagues reported for the first time about the protection efficacy but without lung viral titers upon 5J8 treatment (Krause et al. JV. 2011). Therefore, we rephrased the sentence in a fact description instead of an unexpected tone, "Mice treated with 5J8 showed a relatively low survival rate of 60%, despite of low virus titers in the lung following MA-CA4/2009 (pdm H1N1) challenge." (Page 13, line 269)

Comment 15: Page 13, lines 283-284. "C12H5 at a concentration of 0.1 mg/mL inhibits the egress of all above-mentioned viruses from the infected cells". Again, seems to be an overstatement of the data. There appears to be no egress inhibition of FL/2004.

Response: We have revised the sentence as: “C12H5 at a concentration of 0.1 mg/mL inhibits the egress of all above-mentioned viruses from the infected cells except for FL/2006 strain.” (Page 13, line 286)

Comment 16: Page 14, lines 290-291. "Furthermore, C12H5 can effectively block virus egress, as tested with five H1 and two H5 strains. Seems to be misleading- C12H5 doesn't block FL/2004 egress.

Response: We have revised the sentence as: “Furthermore, C12H5 can effectively block virus egress, as tested with five H1 and two H5 strains but FL/2006 strain.” (Page 14, line 293)

Comment 17: Page 14, lines 305-306, RMSD of 0.526 Å. Please round to 0.5 Å as three decimal points are not meaningful.

Response: Corrected. (Page 15, line 309)

Comment 18: Page 15, lines 313-314. "12H5 indeed targets the RBS of different HAs". Do you mean different protomers within the same HA rather than different HAs?

Response: Yes, we have rephased the sentence as "12H5 indeed targets the RBS of different protomers within the same HA and could occupy all three potential binding sites." (Page 15, line 317)

Comment 19: Page 15, line 313. Please replace "cyro-EM" with "cryo-EM"

Response: Corrected. (Page 15, line 317)

Comment 20: Page 15, line 318. Please replace "Table. S6" with "Table S6".

Response: Corrected. (Page 15, line 322)

Comment 21: Page 16, line 339. Please replace "salt-bridging" with "salt bridges".

Response: Corrected. (Page 16, line 345)

Comment 22: Page 16, line 341. Please replace "bonding" with "bond".

Response: Corrected. (Page 16, line 345)

Comment 23: Page 16, line 342. Please replace "bonding" with "bond".

Response: Corrected. (Page 16, line 347)

Comment 24: Page 16, line 343. Please replace "binding" with "interactions".

Response: Corrected. (Page 16, line 349)

Comment 25: Page 17, line 353. Please replace "sites on" with "site".

Response: Corrected. (Page 17, line 358)

Comment 26: Page 17, line 354. "interaction mimics the interaction of Sia-1". There is some mimicry is H bonds but I would say 'mimics to some extent'.

Response: We agree the comment of reviewer. We have rephased the sentence as: "this interaction mimics the interaction of Sia-1 with the 130- loop to some extent." (Page 17, line 359)

Comment 27: Page 17, line 356. Please replace "GlaNac" with "GalNac"?

Response: Corrected. (Page 17, line 362)

Comment 28: Page 17, line 357. Please replace "meditates" with "mediates".

Response: Corrected. (Page 17, line 363)

Comment 29: Page 19, line 402. "strong" I don't think there are any data on strength.

Response: As suggested, we have deleted the "strong." (Page 19, line 406)

Comment 30: Page 19, line 404. Please replace "mutates" with "mutated".

Response: Corrected. (Page 19, line 407)

Comment 31: Page 19, lines 409-410. "Combined, these residues have an average conservation of 90% across all H1N1 isolates". Does this include the 1918 and other early isolates. With this high degree of conservation, does the antibody not bind 1918 and other early H1N1 viruses?

Response: Yes, the conservation analysis did include the 1918 and other early isolates. But we have not done binding assay with the 1918 strain and other early isolates, as they are not available in our lab.

Comment 32: Page 21, line 443. Please replace "a known" with "the known".

Response: Corrected. (Page 21, line 446)

Comment 33: Page 22, line 477. Please replace "seem relatively" with "seem to have relatively".

Response: Corrected. (Page 22, line 480)

Comment 34: Page 22, lines 478-480. "reciprocal to strain-specific neutralizing antibodies must be redesigned by means of immune focusing during broadly protective immunogen design^{11,12,31}". Please rephrase this whole sentence.

Response: As suggested, the sentence was rephrased as: "These results are consistent with the findings in the literature that broad neutralization epitopes seem to have relatively immunodominance in host immunity than strain-specific ones, and should be rationally designed for more immunogenic in broadly protection by means of immune focusing." (Page 22, line 478)

Comment 35: Page 24, lines 510-511. "CH65 could only neutralize seasonal influenza H1N1 viruses before 2009". What about 1918 and other H1N1 arising from that pandemic?

Response: Sorry for the misleading description. Actually, the CH65 could neutralize the seasonal influenza H1N1 strains from 1993 to 2006 but not 2009 pandemic strains in the previous study (Whittle et al. PNAS. 2011). However, Hong et al. reported CH65 failed in binding to A/South Carolina/1/1918 strains HA (Hong et al. Journal of Virology. 20113). Thus, we have limited the neutralization breadth in our testing panel, "CH65 could neutralize seasonal influenza H1N1 viruses before 2009 in the testing virus panel". (Page 24, line 512)

Comment 36: Page 25, line 535. Please insert ',' before we

Response: Corrected. (Page 25, line 538)

Comment 37: Page 26, line 557. Please delete "190". Please replace "mutations from" with "mutations at residue 190 from".

Response: Corrected. (Page 26, line 558)

Comment 37: Please 26, lines 561-562. "and bear the mutations as related to receptor binding transition". ????

Response: We have rephrased the sentence as: "12H5 can overcome antigenic variation among H1 and H5 subtypes and bear the mutations as related to receptor binding transition from human to avian host." (Page 26, line 562)

Comment 38: Page 27, line 581. Please replace "stain" with "strain".

Response: Corrected. (Page 27, line 583)

Comment 39: Page 37, line 800. Please replace "murmuring" with "numbering".

Response: Corrected. (Page 37, line 797)

Comment 40: Page 38, lines 816-817. "HA was mixed with 12H5 Fab in a molar ratio of 1:1.2". Is this the molar ratio to HA protomer and not to HA trimer?

Response: Yes, we have revised the sentence as "HA was mixed with 12H5 Fab in a molar ratio of 1:1.2 (HA protomer to Fab)." (Page 38, line 814)

Comment 41: Page 39, line 837. Please delete "JV".

Response: Corrected. (Page 38, line 833)

Comment 42: Page 39, line 844. Please replace "Phenixto" with "Phenix to".

Response: Corrected. (Page 39, line 841)

Comment 43: Page 48, line 1056, Reference 24. Please abbreviate journal reference.

Response: Corrected. (Page 37, line 800)

Comment 44: Page 57, line 1197. Fig. 5 title. "Structure of 12H5:HAhrCA complex" could be changed to "Structures of 12H5:H1 HA complexes", as the section is about crystal and EM studies of 12H5 to two different HAs: HAhrCA and BJ/1995.

Response: Corrected. (Page 56, line 1224)

Comment 45: Page 60, Fig. 7, line 1247. Please replace "Tale" with "Table".

Response: Corrected. (Page 59, line 1282)

Supplementary Materials

Page 25, Table S6. Please truncate to no more than one decimal point for the BSA.

Response: Corrected.

Reviewer #2 (Remarks to the Author):

In the present study, Li and colleagues, isolated and characterized a murine antibody 12H5 that cross neutralizes H1N1 and H5N1 viruses, both belonging to group 1 influenza viruses.

The authors showed that the mAb engages the receptor binding site (RBS) of both H1 and H5 hemagglutinins and that it is insensitive to the Asp->Glu amino-acid substitution at position 190 which explains, in part, the broad cross-reactivity of this mAb.

While this is a detailed characterization of an interesting antibody, it is simply that – an interesting mouse antibody which seems to have been serendipitously isolated (sequential immunization with seasonal H1N1 viruses then simply screening of a lot of hybridoma clones). The study does not teach us anything new about the human immune system, nor about strategies of how to induce this class of antibodies in animal

and in humans.

Response: We thank the reviewer for the encouraging comments and pointing out some flaws in our study. Although 12H5 is a murine antibody, which could cross-neutralize H1 and H5 by targeting to HA head region and was never reported in the literature, we believe that it will cast some beneficial lights to other studies regarding to human immune system against highly varied flu and vaccine design for broadly protection. Interestingly, the antibody is produced in mice by the immunization of solely H1 viruses but could cross cover H5 breadth, suggesting that 12H5-like antibody might be able to appear in humans who received multiple vaccinations of annual H1 vaccines. In our ongoing project, we are trying to screen C12H5-like antibodies in peripheral blood mononuclear cells (PBMC) from the subjects with multiple H1 strains vaccinated or natively infected, where co-readout from inflorescence reporters labelled to wildtype H1 HA and 12H5-silenced HA mutants would help to identify this category of antibodies. As to broadly vaccine design, we have some preliminary results that the evicted partial 12H5 epitope showed mild immunogenic while conjugated to KLH carrier, which could serve as the start model and afford a further exquisite design by other protein design techniques. Nevertheless, how to relate a murine epitope to human one in vaccine design is far from reach in immunology, but is a useful and informative avenue thanks to the easy-manipulated mouse model.

Major:

Comment 1: The X-ray data processing and refinement statistics are inconsistent in the 2 tables provided. For example, the PDB deposition report shows R_w/R_f of 26.6/28.9 and the table S1 of the MS 26.9/29.2. Other inconsistencies like data resolution range are also noted.

Response: We have corrected the data in the updated Table S3.

Comment 2: It is surprising that the cryo-EM reconstruction was achieved to only 7Å of nominal resolution with over 300k particles and pixel size 1.12Å. The methods are inadequately described, and it is therefore impossible for me to judge what might have been the limiting factor in the cryo-EM data processing. I dare say that with more rigorous data processing (for ex. trying C3 symmetry, 3D classification and focused refinement these data should yield a 4Å reconstruction (or better).

The MS is also lacking the data processing pipeline, estimates of local resolution and model to volume fit figures. Additionally, cryoSPARC is misspelled in the methods.

Response: The initial cryo-EM reconstruction was obtained at only 7Å resolution by a FEI Falcon2 camera lack of electron counting function. We have resolved the cryo-EM structure by new data collection using our recently updated Gatan K3 detector. The cryo-EM reconstruction finally achieved 3.14Å resolution density map of immune complex (the trimeric HA^{BJ} bound by three 12H5 Fabs) and generated an atomic model for the 12H5:HA^{BJ} complex. We have updated the cryo-EM analysis and detailed the method in our revised manuscript (please see the revised **Fig. 5, Fig. S8, Table S4** and description in Page 15, lines 312-318).

Comment 3: A more detailed structural comparison with 5J8 and CH67 (H1 specific) and FLD21.140 and AVFlulg03 (H5 specific) is warranted.

Response: As suggested, we have added the structural comparison of 12H5 to 5J8 and CH67 (H1 specific) and FLD21.140 and AVFlulg03 (H5 specific) in the updated Fig. 3 and the corresponding description. (Page 16, line 332-338)

Minor:

Comment 4: Why are Supplementary figures called out first? It would be more informative to re-work the main Figure 1 and include the protection data. SPR data can be summarized in a table.

Response: As suggested, we have updated the Figure 1.

Comment 5: CL6649 is the name of the clonal lineage, the mAb used is Ab6649.

Response: Corrected. (Page 37, line 800)

Reviewer comments, second round review

Reviewer #1 (Remarks to the Author):

The authors have taken solid steps to address the questions I raised in the first round of review. I now believe that the manuscript is suitable for publication in Nat. Comm. and that it will be a impactful contribution to the field.

Reviewer #2 (Remarks to the Author):

The revised version could be accepted for publication.

Reviewer comments, second round review

Reviewer #1 (Remarks to the Author):

The authors have adequately addressed the comments from the previous review by this reviewer. Some additional comments are provided below:

Line 344, "salt bridges linkages" to "salt bridges"

Line 1229 and 1232, Fig.4 legend, "influenza A" to "influenza A virus"

Fig. 5 & Fig. 6 legends and throughout text: "PDB no." or "PDB accession no." to "PDB code"

Line 1244, Fig. 5 legend, "of complex with the" to "trimer from"

Line 1245, Fig. 5 legend, "HA trimer with 3.15 Å resolution" to "at 3.15 Å resolution"

Line 1259, Fig. 5 legend, "crystal of structure" to "crystal structures"

Line 1262, Fig. 5 legend, "in complex with HA" to "in complex with HAs"

Line 1267, Fig. 5 legend, "footprint of the four H1N1 head antibodies" to "footprints of the three HA head antibodies"

Line 1272, Fig. 6 legend, "salt-bridging linkages" to "salt bridges"

Line 1277, Fig. 6 legend, "Hydrogen bonding" to "Hydrogen bond interactions"

Reviewer #2 (Remarks to the Author):

This is a markedly improved manuscript. I have no further edits.

Response to Reviewer Comments on the manuscript [NCOMMS-22-11224A]:

We thank the two reviewers for recognizing the merit of our work and for their suggestions to improve our manuscript. We have fully addressed the comments with appropriate additional experiments and analyses. To facilitate the navigation of this document, we have copied the reviewers' comments verbatim in blue and typed our responses in **black**, and some figures related to their corresponding comments are copied here as well.

Reviewer 1 (Remarks to the Author)

The authors have adequately addressed the comments from the previous review by this reviewer. Some additional comments are provided below:

Comment 1: Line 344, "salt bridges linkages" to "salt bridges"

Response: Corrected.

Comment 2: Line 1229 and 1232, Fig.4 legend, "influenza A" to "influenza A virus"

Response: Corrected.

Comment 3: Fig. 5 & Fig. 6 legends and throughout text: "PDB no." or "PDB accession no." to "PDB code"

Response: Corrected.

Comment 4: Line 1244, Fig. 5 legend, "of complex with the" to "trimer from"

Response: Corrected.

Comment 5: Line 1245, Fig. 5 legend, "HA trimer with 3.15 Å resolution" to "at 3.15 Å resolution"

Response: Corrected.

Comment 6: Line 1259, Fig. 5 legend, "crystal of structure" to "crystal structures"

Response: Corrected.

Comment 7: Line 1262, Fig. 5 legend, "in complex with HA" to "in complex with HAs"

Response: Corrected.

Comment 8: Line 1267, Fig. 5 legend, "footprint of the four H1N1 head antibodies" to "footprints of the three HA head antibodies"

Response: Corrected.

Comment 9: Line 1272, Fig. 6 legend, "salt-bridging linkages" to "salt bridges"

Response: Corrected.

Comment 10: Line 1277, Fig. 6 legend, "Hydrogen bonding" to "Hydrogen bond interactions"

Response: Corrected.

Reviewer #2 (Remarks to the Author):

This is a markedly improved manuscript. I have no further edits.